# Polymorphism and evolutionary origins of accessory chromosomes in the basidiomycete *Tremella fuciformis*

Jinxiang Zhang[1,6], Qianwen Tong[1,6], Fengjiao Lin[1,6], Xingcai An[2], Haichen Huang[1], Hualian Chen[1], Jingjing Ye[1], Huaizhen Xu[1], Xiaomeng Lv[1], Zhiwen Lv[3], Fangting Zeng[1], Tuo Zhang[1], Xiaoping Wu[1], Baogui Xie ⓘ[1] ✉, Ray Ming ⓘ[4,5] ✉ & Youjin Deng ⓘ[1] ✉

Accessory chromosomes are non-essential for growth but poorly characterized in basidiomycetes, unlike in Ascomycota. Here, we report whole-genome sequencing of 16 strains of the basidiomycete *Tremella fuciformis* (silver ear fungus), generating 27 complete haplotypes (5 monokaryons and 11 dikaryons, each contributing two distinct haplotypes). Genome size varied by over one-third, driven by accessory chromosomes and repetitive sequences in core chromosomes (essential for basic biology). Each strain harbored 8-10 core chromosomes (polymorphic via fusion/fission) and 2-10 accessory chromosomes (total 108), whose distribution reflects phylogeny and symbiotic specificity with the ascomycete *Annulohypoxylon stygium*. Accessory chromosomes are small, transposon-rich, gene-poor, and exhibit higher sequence similarity but more diverse structural variations than core chromosomes, with few shared genes across phylogenetic branches. Both chromosome types show frequent copy number variation during cell type transformation. Most accessory chromosome genes lack homologs in core chromosomes or existing gene databases. Our study reveals basidiomycete accessory chromosome diversity, suggesting an origin from unexplored species pre-dating *T. fuciformis* speciation.

With the advancement of next-generation sequencing technologies and telomere-to-telomere (T2T) genome assembly techniques, an increasing number of complete fungal genomes have been decoded[1,2]. These developments have facilitated a deeper understanding of distinctive genomic characteristics exhibited by some fungi, such as the two-speed genome phenomenon[3]. The genomic structure comprises core genome and accessory genome. The core genome is defined as the set of core chromosomes (CCs) shared by all isolates, encoding all essential functions required for survival under normal growth conditions[4–6]. Accessory genome refers to the parts of fungal genomes that may consist of compartments from CCs or entirely accessory chromosomes (ACs), which are also known as dispensable, lineage-specific, supernumerary, and B chromosomes[7]. ACs are prevalent in animals, plants, and microorganisms[8]. In fungi, research on accessory chromosomes has primarily focused on pathogens within the Ascomycota phylum, such as species of *Fusarium*, *Zymoseptoria*,

¹College of Life Science, Fujian Agriculture and Forestry University, Fuzhou, Fujian, China. ²College of Agriculture, Yunnan University, Kunming, Yunnan, China. ³State Key Laboratory of Microbial Diversity and Innovative Utilization, Institute of Microbiology, Chinese Academy of Sciences, Beijing, China. ⁴Department of Plant Biology, University of Illinois at Urbana-Champaign, 1201 W. Gregory Drive, Urbana, IL, USA. ⁵College of Future Technology, Fujian Agriculture and Forestry University, Fuzhou, China. ⁶These authors contributed equally: Jinxiang Zhang, Qianwen Tong, Fengjiao Lin. ✉e-mail: mrcfafu@163.com; rayming@illinois.edu; dengyoujin1980@163.com

*Metarhizium*, and *Alternaria*[7,9–11]. ACs in *Fusarium*, *Metarhizium*, and *Alternaria* contain virulence factors, including effector proteins, secondary metabolite biosynthesis clusters, or other metabolic gene clusters[4,11,12]. However, accessory chromosomes in Basidiomycetes are rarely reported. To date, only a handful of species-including *Sporisorium scitamineum*[13] and *T. fuciformis*[14] have been confirmed to harbor them, verifying their presence in this phylum. Documented cases are far fewer than in Ascomycetes, underscoring the need for further exploration in understudied taxa.

Fungal ACs are characterized by distinct sequence features that, while not entirely unique, set them apart from core chromosomes. Typically smaller, ranging from 0.2 to 3.5 Mb[15], ACs are enriched with transposons or repetitive sequences and exhibit lower gene densities and distinct codon usage compared to CCs[16]. These non-essential chromosomes evolve rapidly and often harbor genes related to pathogenicity and adaptation[17], highlighting their unique role in the organism's biology. Predominantly heterochromatic, ACs are marked by significant enrichment of histone modifications such as H3K9me3 and H3K27me3[18]. H3K27me3 is enriched across nearly the entire chromosome and is linked to the instability of accessory chromosomes, whereas H3K9me3 is mainly associated with repetitive sequences and contributes to genome stability[18–20]. While CCs can sometimes include compartments with low gene density, unique heterochromatin structures, and high repeat content, these features are less pronounced and widespread than in ACs. Fungal accessory chromosomes can sometimes be identified using (epi-)genomic characteristics, though presence/absence polymorphism remains a key criterion for their identification[19,21].

The origin of fungal accessory chromosomes remains unresolved, with two main non-exclusive hypotheses: they may originate from core chromosomes and diverge or degenerate over time[19], or they may be acquired through horizontal chromosome transfer from different lineages or species[22]. In some species, the high similarity in sequence composition between core and accessory chromosomes suggests that accessory chromosomes may have originated from core chromosomes. Recent macrosynteny analyses in the rice blast pathogen *Magnaporthe oryzae* have traced the emergence of accessory minichromosomes through structural rearrangements and segmental duplication of core chromosomes[23]. In *Zymoseptoria tritici*, genome sequencing of meiotic progenies has revealed that the accessory chromosomes primarily originated from ancient core chromosomes through a degeneration process involving the breakage-fusion-bridge cycle[24]. The horizontal chromosome transfer model is also supported by some evidence. For instance, differences in codon usage between core and accessory chromosomes suggest this possibility[25]. Phylogenetic analysis indicates that the accessory chromosomes of *Fusarium oxysporum* f.sp. *lycopersici* have a different origin than the core chromosomes[4]. Experimental evidence has demonstrated mechanisms that enable the horizontal transfer of entire chromosomes between distinct lineages in species such as *F. oxysporum*[21,26], *Alternaria* spp[27]., *Metarhizium robertsii*, and *Colletotrichum gloeosporioides*[28].

*Tremella fuciformis*, commonly known as snow fungus or silver ear fungus, is a unique and widespread fungus that thrives in tropical regions, often found on the dead branches of broadleaf trees. It belongs to the order Tremellales and the family Tremellaceae[29]. The fruiting body formation of *T. fuciformis* is extremely dependent on its companion fungi *Annulohypoxylon stygium* in large-scale cultivation and natural environment[30,31]. Specifically, *A. stygium* maintains a close association with *T. fuciformis* and functions as a key nutritional provider, supplying essential nutrients to support the growth and development of the latter. *T. fuciformis* is notable for its dimorphic nature, existing either as yeast-like cells that reproduce asexually through budding or as filamentous cells that grow by apical extension[14]. Similar to *Cryptococcus neoformans*[32], the haploid phase in the life cycle of *T. fuciformis* exists exclusively in the yeast form and cannot transition to

the mycelial form, thus being unable to interact with *A. stygium* hyphae as unicells or form fruiting bodies; in contrast, the heterokaryotic phase is dimorphic, enabling such hyphal interactions and fruiting body development. In previous studies, three accessory chromosomes have been identified in the genome of *T. fuciformis* Tr01, exhibiting small size (<1 Mb), low gene density, and a high proportion of repetitive sequences. They may be absent in their basidiospores or exist in multiple copies[14]. However, their distribution within cells, genetic origin, functions, and impact on the growth and reproduction of *T. fuciformis* remain unknown.

In this study, we intend to conduct whole-genome sequencing assembly and comparative genomic analysis of 16 *T. fuciformis* strains from different regions. Our aim is to explore the diversity of accessory chromosomes within *T. fuciformis*, explore their shared sequence characteristics, and investigate their distribution and stability within cells. Additionally, we will analyze their genetic origins and impact on the host cell genome. This research will enrich our understanding of the genomic diversity of *T. fuciformis* and provide deeper insights into the characteristics of accessory chromosomes in basidiomycete fungi.

## Results

### Haploid genome assemblies of 16 *T. fuciformis* strains

A total of 16 *T. fuciformis* strains were collected from the provinces of Fujian, Guangdong, Sichuan, and Yunnan, China. Among these, two principal cultivated varieties (Tr01 and Tr21), while the remaining 14 are wild isolates. The strain collection comprises 11 heterokaryotic strains and 5 monokaryotic strains (Supplementary Table 1). Among these, 5 heterokaryotic strains were found to harbor two distinct ITS sequences each, whereas the remaining 11 strains contained only one ITS sequence. Phylogenetic analysis based on 21 target ITS sequences and 10 downloaded reference ITS sequences yielded a tree that clustered all tested strains into three distinct clades (Supplementary Fig. 1), with all clades belonging to *T. fuciformis*. NCBI BLAST analysis of the ITS sequence indicated that the isolates are more closely related to the known species *T. fuciformis*. Additionally, their fruiting bodies exhibit the same morphological characteristics as those of *T. fuciformis* (Supplementary Fig. 2). *T. fuciformis* strains within the same cluster can mutually pair with their associated fungus, *A. stygium*, whereas those from different clusters are unable to pair (Supplementary Fig. 3). It is suggested that the 16 strains from these three clusters each belong to a distinct subspecies within the *T. fuciformis* species.

The PacBio Sequel IIe system was utilized to conduct whole-genome sequencing on 15 *T. fuciformis* strains, excluding Tr01 (Cluster III) with its known genome, using the circular consensus sequencing (CCS) and continuous long read (CLR) mode. The sequencing process produced between 40,120 and 575,272 HiFi reads, with an N50 length ranging from 12 Kb to 23 Kb. The total sequencing output was between 5.5 Gb and 10.5 Gb, resulting in a genome coverage of 100× to 450× (Supplementary Table 2). ~40 Gb of data was generated from the Hi-C sequencing of both NDB and TF2206. HiFiasm with Hi-C mode was used to assemble HiFi reads of NDB into 77 contigs. Using the Tr01 genome as a reference, these NDB contigs were further assembled into 19 telomere-to-telomere gapless chromosomes, which were assigned into two sets of haploid genomes by NuclearPhaser (Supplementary Fig. 4). Similarly, the complete heterokaryotic genome of TF2206 was generated from 50 contigs, comprising 13 pairs of telomere-to-telomere, gapless chromosomes. Hi-C interaction heatmaps produced by Hi-C Pro validated the assembled genomic architecture (Supplementary Fig. 5). Using the genomes of TF2206, NDB, and Tr01 as references for Clusters I, II, and III, respectively, the HiFi reads of the other 13 strains were assembled into either one complete haploid genome for monokaryotic strains or two sets of complete haploid genomes for heterokaryotic strains. In total, we have successfully obtained 27 complete haploid genomes, including two previously established from

the Tr01 strain (designated Tr01A and Tr01B, corresponding to its two distinct haplotypes), to support subsequent research.

## Divergence in genome size and chromosome number across genomic landscapes

Pan-genome analysis of the 27 haploid genomes identified 14,567 gene families, including 3549 core genes (24.3% of the total), 9638 accessory genes (66.2%), and 1380 singletons (9.7%). Using 3549 core genes from the pangenome, a phylogenetic tree was constructed (Fig. 1a, column 1). This tree reveals three major clusters, which are consistent with the phylogenetic tree based on ITS sequences and largely align with the three clades in the mitochondrial genome-based phylogenetic tree-with the only exception being strain T0053, which was reclassified from Cluster III to Cluster II (Supplementary Fig. 6).

The 27 haploid genomes of *T. fuciformis* exhibit significant size variation (Fig.1a, column 3). The largest genome is from strain NDB hapA, measuring 32.48 Mb, while the smallest is from strain TF4-1 hapA, measuring 23.83 Mb, indicating a size difference of over one-third. Additionally, there is substantial variation in their chromosome numbers, ranging from 11 to 19 (Fig. 1a, column 2). Core chromosomes, identified as highly conserved and shared by all *T. fuciformis* isolates (Supplementary Fig. 7), may vary in number across different strains due to genetic events such as chromosomal recombination-for example, 8 in Tr01 HapA and 9 in Tr01 HapB[14]. Consistent with this observation, we found that the 27 haploid genomes harbor 8 to 10 core chromosomes, whereas the number of accessory chromosomes ranges from 2 to 10. The size of the accessory chromosomes ranges from 1.4 Mb to 5.2 Mb and shows a positive correlation with the total genome size ($R^2 = 0.81$) (Fig.1b top). In addition to the accessory chromosomes, we also found significant differences in the core genome sizes among the haploid genomes. By dividing the core chromosomes into repetitive and non-repetitive sequence regions, we discovered that the size of the repetitive sequence regions shows a strong positive correlation with the total genome size ($R^2 = 0.90$) (Fig.1b bottom), being the main source of variation in the core genome size. In contrast, within the same heterokaryotic strain, the two sets of chromosomes exhibit fewer differences in both number and size, with the variation in genome size being less than 4%. There are significant differences in genome sizes among the three clusters. The average size of the six haploid genomes in Cluster II is 31.79 Mb, which is larger than the average size of the twelve genomes in Cluster III, measured at 27.33 Mb ($P < 0.0001$). Furthermore, the average genome size of Cluster III is significantly greater than that of the nine genomes in Cluster I, which have an average size of 24.02 Mb ($P < 0.0001$).

Three scenarios contribute to the differences in the number of core chromosomes among these *T. fuciformis* species, and even between the two haploid genomes within the same dikaryotic strain. In strain TF2206, Chr01-1 exhibited strong collinearity with the 5′ end of Tr01's Chr01 chromosome, while Chr01-2 showed similar collinearity with the 3′ end. The junction of the collinear sequences on Tr01's Chr01 chromosome was located at Chr01:2,260,933-2,275,108, identified as a 14,175 bp DNA transposon T1 (Fig.1c). Hi-C interaction heatmaps revealed that while Tr01's Chr01 chromosome was continuous at this junction (Fig.1d top), the 3′ end of Chr01-1 and the 5′ end of Chr01-2 in TF2206 were not connected and contained telomeric interaction hotspots (Fig.1d bottom), indicating accurate chromosome assembly. Tr21 and Tr01 are the only two strains in a branch at the seventh node of the phylogenetic tree, each having only one chromosome (Chr01) in this region, whereas the other strains possess two chromosomes: Chr01-1 and Chr01-2 (Fig.1e). The chromosomal rearrangement was further validated by mapping long HiFi and nanopore reads to the corresponding regions of TF2206 and Tr01 (Supplementary Fig. 8). Similarly, strains in Cluster I and Cluster II possess a single chromosome for Chr02, whereas strains in Cluster III consist of two separate chromosomes (Supplementary Figs. 9 and 10). The third scenario

involves the published Tr01 genome[14], where haplotypeB is similar to haplotypeA in genome size, gene count, and proportion of repetitive sequences. However, chromosomal rearrangements are observed in Chr01, Chr02, and Chr05, along with an additional core chromosome, Chr12B, resulting in a difference in the number of core chromosomes in its heterokaryotic genome.

## Regularity in the distribution of accessory chromosomes

The chromosomes of 27 haploid genomes were aligned with the core chromosomes Chr01-Chr08 of the Tr01 hapA genome, resulting in the eight sets of homologous core chromosome groups, designated as CC01-CC08. Similarly, 108 accessory chromosomes were identified and categorized into 15 groups of homologous chromosomes, named AC01-AC15, based on synteny. The distribution of the 15 sets of accessory chromosomes across the 27 haploid genomes was compared with the phylogenetic tree constructed from the core genes (Fig. 2). The homologous chromosomes of AC01 to AC07 are found in the genomes within Cluster II, with the QSA and QSB genomes lacking AC03, and the NDBA and NDBB genomes lacking AC06. The homologous chromosomes of AC09 to AC12 are located in the genomes of Cluster I, where one or more genomes do not carry AC09, AC10, or AC11. The homologous chromosomes of AC13 and AC14 are present in the genomes of Cluster III. The three largest accessory chromosomes, AC01, AC12, and AC14, are each located on one of the three branches of the phylogenetic tree. All strains in the genomes of each branch carry these chromosomes, except for T0053. Synteny analysis of accessory chromosomes in the QCA and TF104A genomes revealed that their homologous chromosomes AC10 to AC12 are nearly identical (similarity >99.9%, coverage >99.3%; Supplementary Fig. 11), while QCA lacks AC09. All strains in Cluster I, except QC and TF103, harbor AC09, suggesting this accessory chromosome was originally present in the cluster but lost during evolution. The same pattern applies to other chromosome losses. These findings demonstrate that the overall distribution pattern of these 15 accessory chromosomes in the haploid genome aligns with the three branches of the phylogenetic tree. Each branch's heterokaryotic *T. fuciformis* strains could only pair with the associated *A. stygium* strains of that branch and not with *A. stygium* strains from other branches, indicating a specific pairing between *T. fuciformis* and *A. stygium*. It is evident that the distribution pattern of accessory chromosomes matched that of the corresponding associated fungus.

## Common features of accessory chromosomes

By conducting a comparative analysis of the sequence similarities and differences between the core chromosome groups CC01-CC08 and the accessory chromosome groups AC01-AC15, the sequence characteristics of the accessory chromosomes in *T. fuciformis* were summarized. The core chromosomes of the test strains all exceed 1 Mb in size, with the largest being the CC01 homologous chromosome group, averaging 8.45 Mb in length. In contrast, the accessory chromosomes are generally smaller. Except for the AC01, AC12, and AC14 groups, which have average lengths of 2.57 Mb, 1.11 Mb, and 1.24 Mb, respectively, all other accessory chromosomes are less than 1 Mb in length (Fig. 3a). The core chromosomes carry an average of 33.86 to 39.26 genes per 100 Kb sequence. In contrast, the gene density of accessory chromosomes is significantly lower, ranging from 10.29 to 21.71 genes per 100 Kb, with a statistically significant difference ($P$-value = 7.97E-15, Fig. 3b). The average length of genes and individual introns in accessory chromosomes is comparable to that of core chromosomes. However, the average number of introns per gene in accessory chromosomes is significantly lower, with only 2.1 introns per gene, which is notably less than the 3.3 introns per gene in core chromosomes ($P$-value = 7.83E-13, Supplementary Fig. 12). The average proportion of repetitive sequences in accessory chromosomes ranges from 40.16% to 80.98%, significantly higher than that in core chromosomes, which

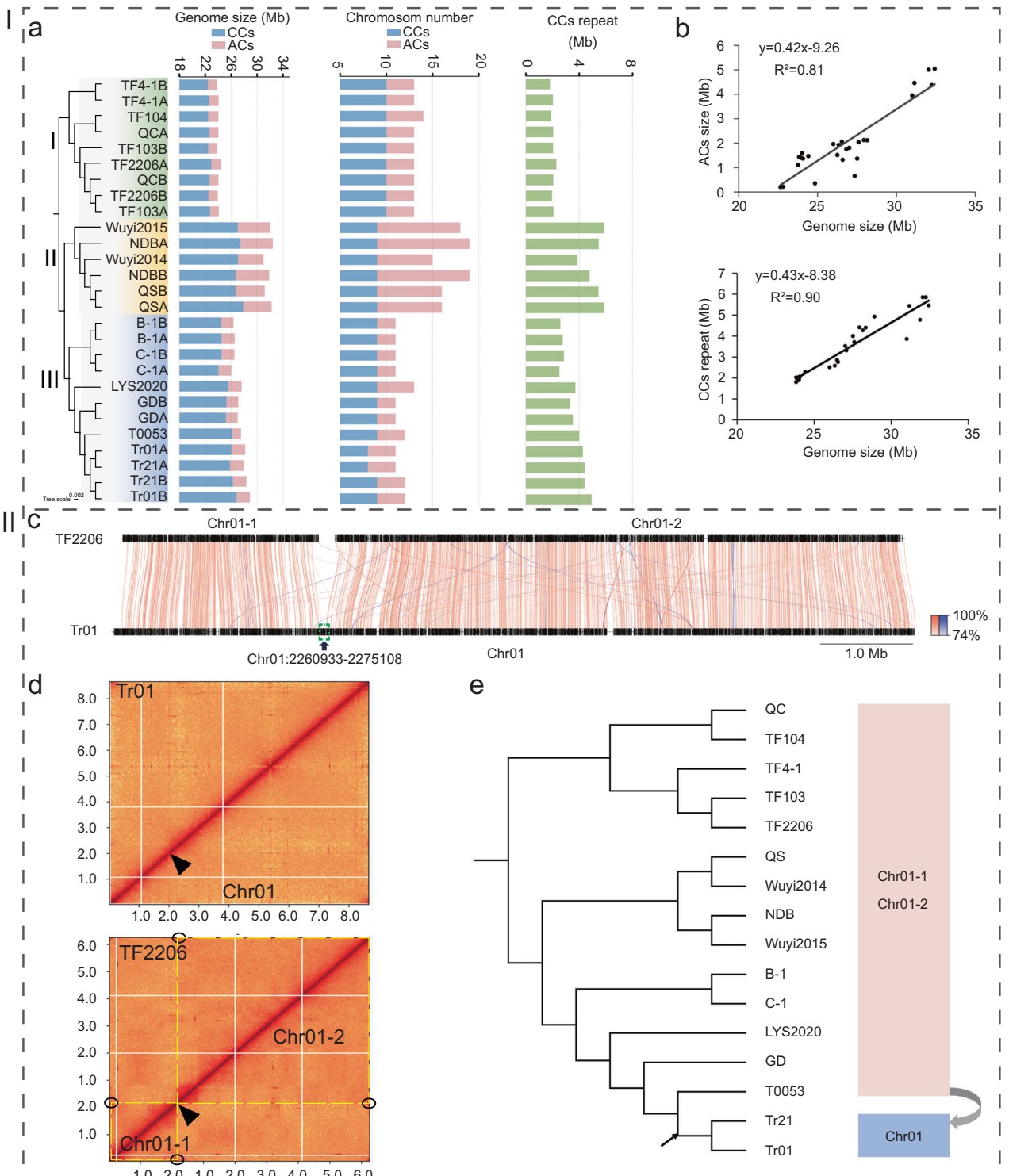

**Fig. 1 | Genomic variation among 27 *T. fuciformis* haploid genomes.** Haploid genomes of monokaryotic strains are named directly after the strain, while those of dikaryotic strains are labeled strain name + A/B to distinguish the two haplotypes. CC, core chromosome; AC, accessory chromosome. **a** Column 1 to column 4 represent phylogenetic tree constructed by core genes, genome size, chromosome number, and repeat sequence length in CC of 27 *T. fuciformis* haploid genomes, respectively. **b** Correlation between genome size and AC size; correlation between genome size and repeat sequence length in CCs. **c** Synteny analysis of Chr01 between Tr01 and TF2206. Green box indicates the boundaries of syntenic regions. **d** Hi-C contact heatmaps of Chr01 (40-Kb resolution) for Tr01 (top) and TF2206 (bottom). Black triangles indicate the boundaries of syntenic regions, while solid black circles mark telomere interaction hotspots. **e** Evolutionary relationships of two types of Chr01 chromosomes based on the phylogenetic tree. Source data are provided as a Source Data file.

ranges from 11.44% to 19.10% (*P*-value = 3.87E-9, Fig. 3c). LTR/Ty3, LTR/Copia, TIR/Mutator, TIR/CACTA, helitron and repeat_fragment are the six most abundant TE types in the *T. fuciformis* genome. Our comparative analysis of their proportional abundances between ACs and CCs revealed that not all TEs are equally distributed between the two chromosome types: relative to CCs, ACs show significantly reduced proportions of TIR/Mutator and TIR/CACTA, while LTR/Ty3 and LTR/Copia are significantly enriched (Supplementary Fig. 13). The average

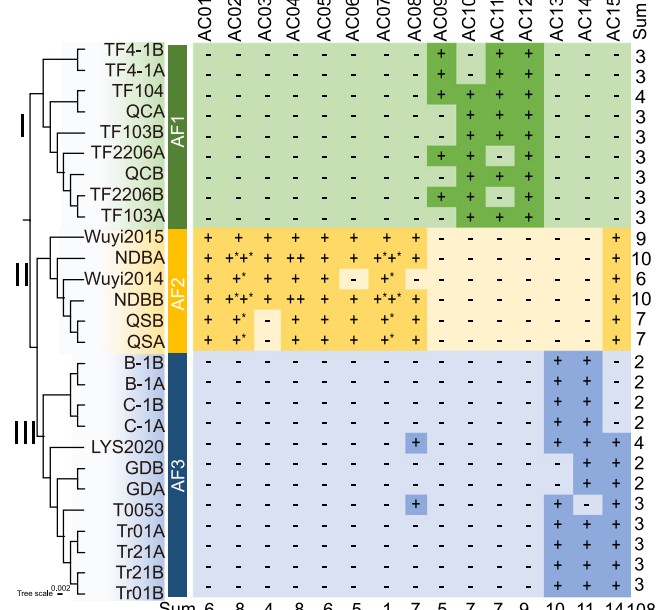

**Fig. 2 | Distribution of 15 groups of accessory chromosomes across 27 haploid genomes.** The left panel shows the phylogenetic tree based on 3,549 core genes from the pangenome. The right panel illustrates the distribution of accessory chromosomes. Minus signs indicate the absence, while plus signs indicate the presence of the corresponding accessory chromosomes. Different branches are highlighted in distinct colors to show the distribution of accessory chromosomes. (*) indicates that the corresponding chromosome in a haploid genome shows partial homology to chromosomes assigned to two accessory chromosome groups. In such cases, the chromosome was counted only once in statistical summaries to avoid double counting.

nucleotide identity (ANI) among homologous core chromosomes in 27 haploid genomes ranged from 86.7% to 88.4%. In contrast, the similarity between homologous accessory chromosomes is higher, with median ANI values ranging from 97.4% to 99.6%, showing a significant difference (*P*-value = 4.01E-21, Fig. 3d).

## The rapid structural variation of accessory chromosomes

Collinearity analysis of homologous accessory chromosomes revealed their rapid structural variations, including inversion, terminal chromosome loss, insertion-deletion, and chromosome fusion-splitting. In the phylogenetic tree, the NDBA genome is closest to Wuyi2015. In Wuyi2015, the latter 1.0 Mb of the AC01 chromosome, from position 1,672,381 to the end, has undergone an inversion (Fig. 4a). Moreover, several smaller inversions are present throughout the AC01 chromosome. Compared to Wuyi2015, the first half of the AC08 chromosome in NDBA (positions 1 to 41,628 and 46,673 to 72,109) has experienced a deletion, accounting for one fifth of its total length (Fig. 4b). The sequence similarity between the homologous AC14 chromosomes of the A and B haplotypes in the dikaryotic B-1 strain is as high as 99.5%. However, there are four insertion-deletion segments exceeding 50 Kb and one inversion of 107 Kb between them (Fig. 4c). The AC14 of GDA is collinear with both AC13 and AC14 of Tr21A, indicating a chromosome fusion-splitting event (Fig. 4d). Additionally, the latter half of the AC13 chromosome in Tr21A (from position 231,466 to the end) has been lost, while the middle section of the AC14 chromosome in GDA (from position 356,504 to 531,280) has also been lost.

Eight Starship giant transposons (41,942-605,797 bp) were identified across 6 genomes, with 3 localized to core chromosomes and 5 to ACs. Notably, SS04 is positioned at a chromosome fusion site (Fig. 4d), suggesting involvement in fusion events. Further, 6 Starships formed 3 groups on homologous loci of heterokaryotic strains: SS02/SS03 (core Chr06) were identical, while AC-localized pairs (SS06/SS07 on GD

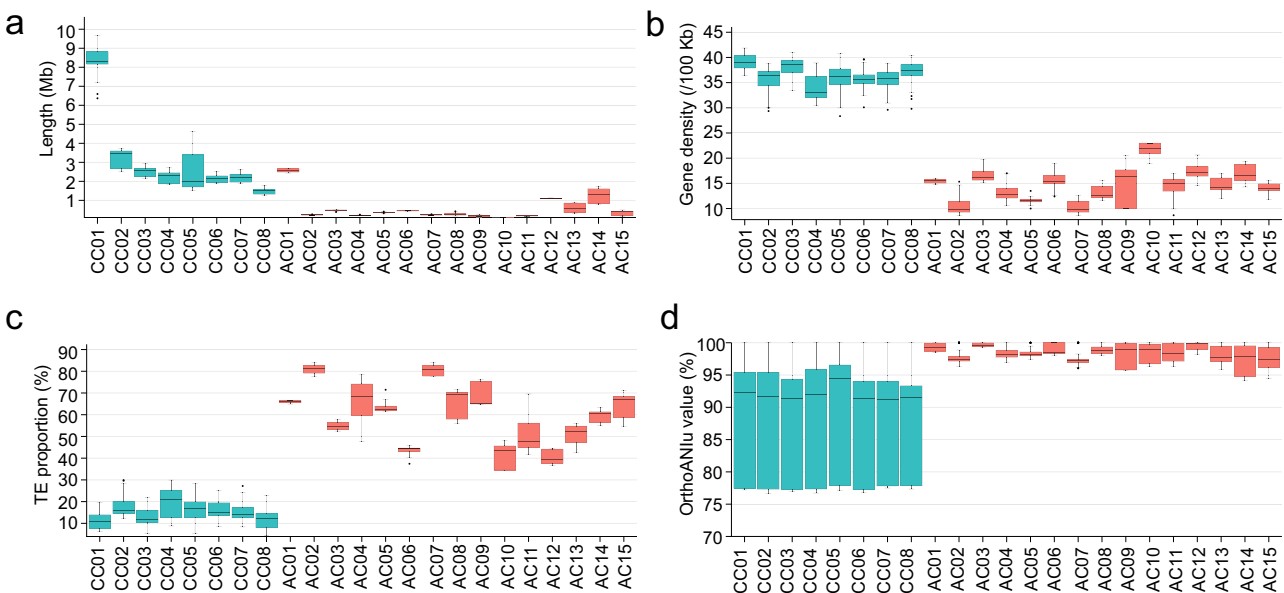

**Fig. 3 | Features of accessory chromosomes.** CC, core chromosome as light blue; AC, accessory chromosome as light yellow. **a** Length distribution among core and accessory chromosomes. **b** Distribution of gene density among core and accessory chromosomes. **c** Distribution of TE proportion among core and accessory chromosomes. **d** Distribution of average nucleotide identity (ANI) among chromosomes within the core and accessory chromosome sets. Box plots show the distribution of chromosome-level features across predefined chromosome groups. Box plots show the distribution of chromosome-level features across predefined chromosome groups. The sample size (n) indicates the number of chromosomes included in each group, and individual chromosomes within each group represent the unit of

study. Accordingly, the box plots primarily reflect overall distributional differences among chromosome groups, rather than assuming complete statistical independence of individual chromosomes. In each box plot, the center line represents the median, the box denotes the interquartile range (25th–75th percentiles), whiskers indicate the full data range (minimum to maximum), and outliers are shown as individual points. All data points represent biological replicates; no technical replicates were used. Statistical definitions, sample sizes, and units of study are provided in Supplementary Table 4. Comparisons between core (CC) and accessory (AC) chromosomes were performed using one-way ANOVA. Source data are provided as a Source Data file.

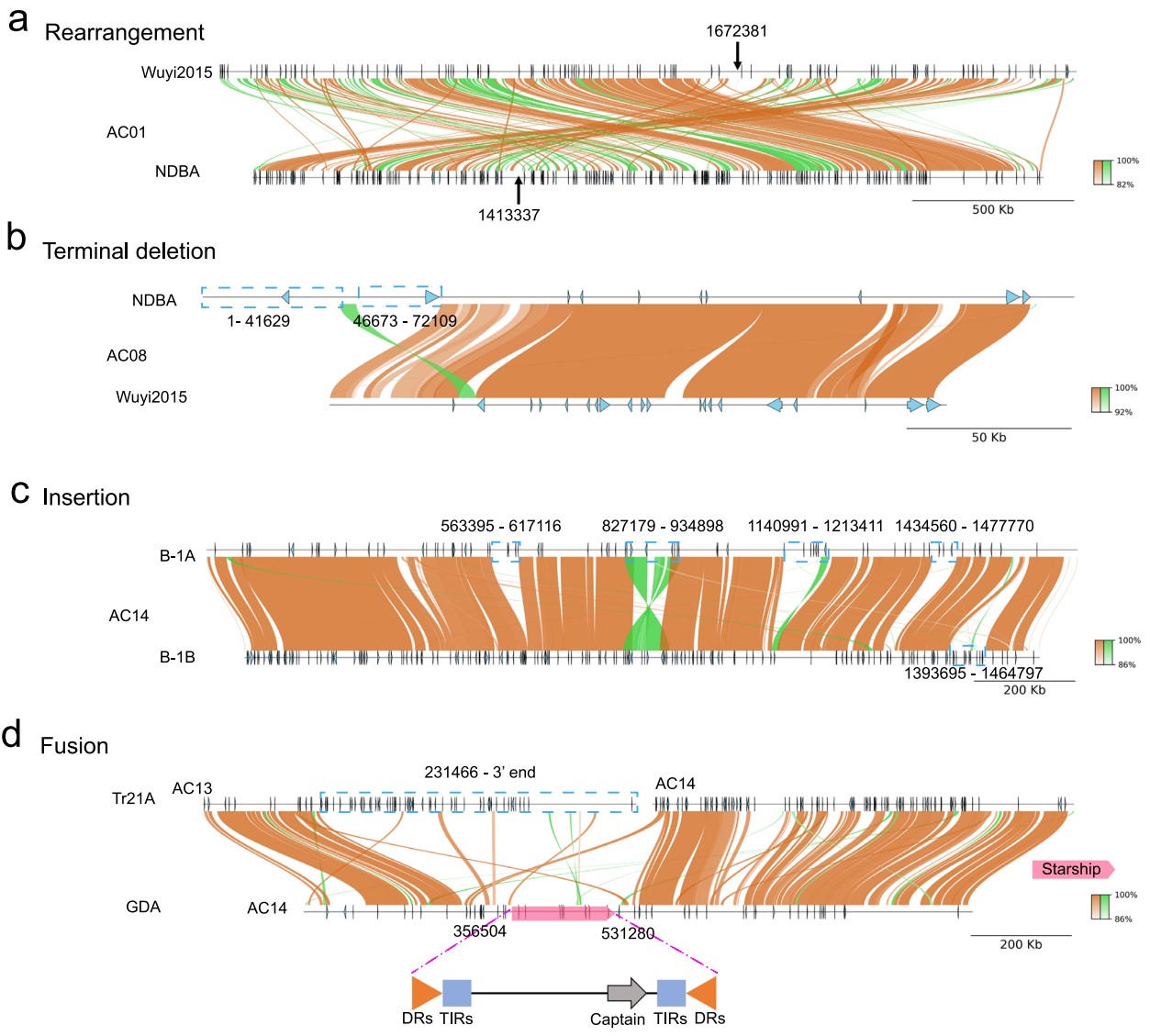

**Fig. 4 | Structural variations in accessory chromosomes. a** Chromosomal rearrangement events between Wuyi2015 and NDBA strains on AC01 chromosome, involving sequences from position 1,672,381 to the terminus. **b** Terminal deletions on AC08 chromosome between NDBA and Wuyi2015 strains, with deleted regions spanning positions 1-41,629 and 46,673-72,109. **c** Insertion events between B-1A and B-1B strains on AC08 chromosome. Inserted sequences are located at positions 563,395-617,116; 1,140,991-1,213,411; 1,434,560-1,477,770; and 1,393,695-1,464,797,

with a concomitant rearrangement observed at 827,179-934,898. **d** Chromosomal fusion events involving the Tr21A region of both AC13 and AC14 chromosomes and the GDA region of AC13 chromosome. This complex variation comprises a terminal deletion from position 231,466 in Tr21A region and an insertion of GDA sequences (356,504-531,280) corresponding to SS04, which contains a typical captain gene, direct repeats (DRs), and terminal inverted repeats (TIRs).

Chr09 and SS04/SS05 on C-1 Chr10) showed structural variations (46 bp and 3048 bp with 4 LTRs, respectively), supporting rapid AC structural variation (Supplementary Table 3 and Supplementary Fig. 14).

## High frequency of chromosomal copy number variation in both core and accessory chromosomes

*T. fuciformis* can exist in either mycelial or yeast form. To illustrate its chromosomal copy number variation (CNV) characteristics, we randomly selected dikaryotic strain TF2206 as a case study and analyzed CNV in its three states. A single yeast-forming colony was expanded in liquid medium for 3 days, then diluted and plated onto agar plates. Ten single colonies were randomly selected for sequencing analysis, and copy number variations were found in six chromosomes (Fig. 5a). In the ten isolated colonies, the copy numbers of chromosomes Chr07, Chr09, and Chr10 consistently varied, being 2.0×, 0.5×, and 0× their original values. For chromosome Chr01-2, only the YY3 colony exhibited

a copy number change to 0.8×. In the case of chromosome Chr05, all colonies except YY3 had a copy number of 1.5×, with YY3 having a copy number of 1.1×. Regarding chromosome Chr11, the copy number in the YY1 colony remained unchanged, while it was 0.9× in the YY8 strain and 0.5× in the other colonies. It can be inferred that the chromosomal copy number of the initial single yeast-forming colony has undergone variation. Specifically, the copy numbers of Chr05, Chr07, Chr09, Chr10, and Chr11 have changed to 1.5×, 2.0×, 0.5×, 0×, and 0.5× their original values, respectively. During the expansion process, three colonies exhibited four instances of variation: Chr01-2 and Chr05 in colony YY3, and Chr11 in colonies YY1 and YY8. We also conducted a chromosomal copy number analysis on 10 mycelial colonies obtained from subculturing the same original parent colony (Fig. 5b). The results showed that the copy number of the core chromosome Chr06 in all colonies had varied, reducing to 0.75×. Additionally, the three accessory chromosomes also exhibited copy number variations, with Chr09 and Chr10 each having two types of copy number variations,

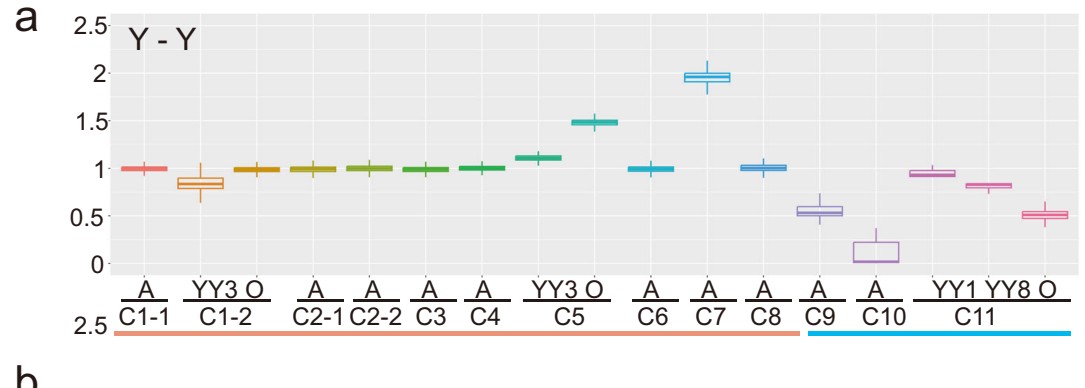

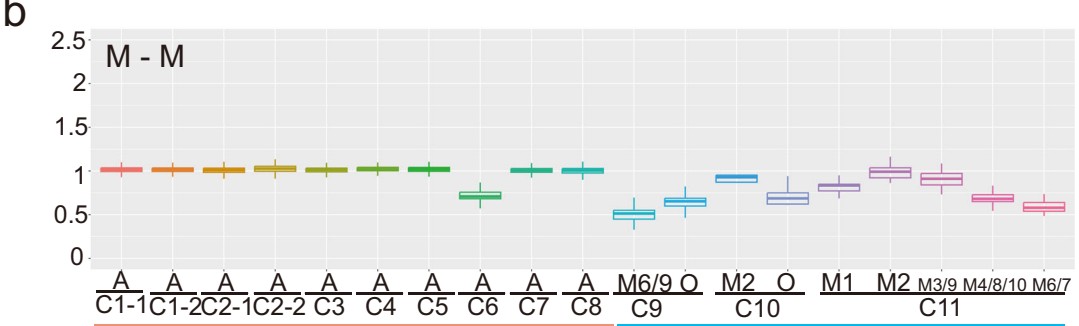

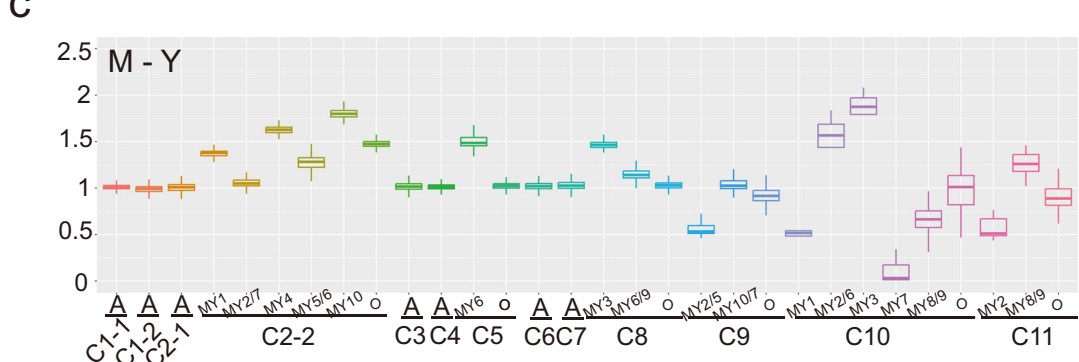

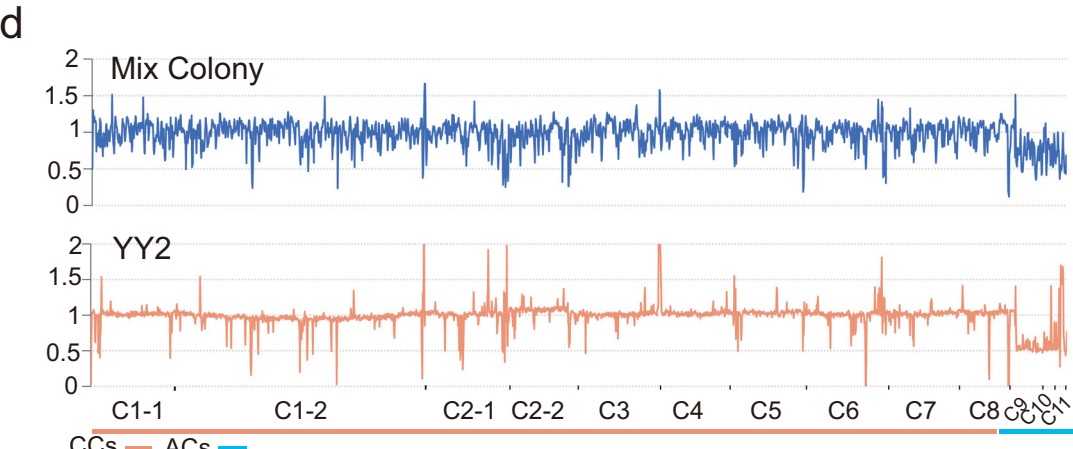

**Fig. 5 | Chromosomal copy number variations in *T. fuciformis* TF2206.** C1-1 to C11 represent the 13 chromosomes of *T. fuciformis* TF2206, with the last three being accessory chromosomes and the others being core chromosomes. On the horizontal axis, Y1-Y10 correspond to 10 single-yeast forming colonies, and M1-M10 correspond to 10 mycelial colonies. A represents all colonies, and O represents other colonies. **a–c** Is box plots of normalized sequencing depth (×), showing 10 single-yeast germination colonies from one colony (**a**), 10 mycelial transfer colonies from one colony (**b**), and 10 single-yeast purified colonies from different mycelium-to-yeast transitions (**c**). **d** Shows the normalized depth distribution curves of Illumina sequencing between a single-yeast forming colony (YY2) and a mixed yeast colony transition. For box plots in panels a-c, each data point represents a non-overlapping 1 Kb genomic window along the chromosome. Definitions of sample size, statistical units, and box plot elements are provided in Supplementary Tables 5–7. Source data are provided as a Source Data file.

while Chr11 had four types. *T. fuciformis* mycelium can be transformed into yeasts, and through streak purification techniques, we repeated the process 10 times, each time obtaining one single-yeast forming colony. We conducted chromosomal copy number analysis using 10 single-yeast forming colonies and one mixed culture before purification. For these single-yeast forming colonies (Fig. 5c), copy number of three core chromosomes has undergone variation, with Chr05 exhibiting one type of copy number variation, Chr08 having two types, and Chr02-2 having five types, all of which show an increase in copy number. Additionally, the three accessory chromosomes also display copy number variations, with Chr09 having two types, Chr11 having three types, and Chr10 having five types. We compared the normalized depth distribution curves of Illumina sequencing between a representative single-yeast forming colony (YY2, with stable genetic background) and a mixed yeast colony transitioning from mycelium (Fig. 5d). The results showed that the YY2 curve was relatively stable across most regions, whereas the curve for the mixed colonies exhibited significant fluctuations. These fluctuations might be attributed to the presence of numerous yeasts with varying sequencing depths within the mixed colony.

### Possible origins of accessory chromosomes

To investigate whether accessory chromosomes (ACs) originated from core chromosomes or external sources, we took AC01 as a representative example. Among the 2340 genes in AC01, 1334 (57.0%) are shared by all six strains, 99 (4.2%) are unique to a single strain (Fig. 6I-a), and only 332 (14.2%) exhibit homology with core chromosome (CC) genes across the six haploid genomes carrying this chromosome (Fig. 6I-b). Of these 332 shared genes, 256 (77.1%) were embedded within TE sequences (Fig. 6I-c and Fig. 6I-d). For AC02 to AC15, the majority of their genes lack counterparts in core chromosomes (Fig. 6I-e), and the limited shared genes are associated with TE activity (Fig. 6I-e). These findings collectively suggest that ACs are not derived from the detachment of CCs but were acquired through horizontal gene transfer from external sources.

Strains of Cluster I, II, and III carry 325, 891, and 590 AC genes respectively, totaling 1806 genes (Fig. 6II-a), among which 1548 are AC-specific genes. Only 102 accessory genes (16.9% on average) are homologous across all three clades, while the number of shared core genes reaches 7862 (accounting for 75.3%, Fig. 6II-a). Among the 1548 AC-specific genes, 1156 (75%) failed to match any homologous sequences or conserved domains, and lacked functional motifs, signal peptides, or transmembrane domains (Fig. 6II-b). This not only indicates that the majority of these genes have not been functionally characterized in previous studies, but also strongly suggests they likely originate from unexplored or understudied species. Of the remaining 392 genes, 232 showed matches to homologous regions, but with unknown functions, 7 lacked homologous matches and encoded proteins bearing signal peptides, leaving only 153 with annotated functions. Among these 153 functionally characterized genes, the most abundant were those encoding motor domains and members of the major facilitator superfamily proteins involved in transmembrane transport of nutrients, energy conversion, and molecular movement (Fig. 6II-c). Notably, 236 of the 392 genes with homologous matches exhibited the highest sequence similarity to species within the Tremellales (Fig. 6II-d).

The proportion of repetitive sequences in ACs varies significantly, ranging from 38% to 89% (Fig. 6III-a). Among these repetitive sequences, RNA transposons are the most abundant (34%–75%). The repetitive sequences in ACs of Cluster I and Cluster III share a higher proportion with those in CCs, while the corresponding proportion for Cluster II ACs is less than 16% (Fig. 6III-b). Taking AC01 as a case study, ACs and CCs contain 3 and 19 intact Type 1 LTR retrotransposons, respectively, among which 2 and 5 have sequence differences between the two terminal LTRs. Sequence analysis of these differentially

terminal LTRs shows that the LTR retrotransposons originally existed in CCs and were first inserted into ACs 500,000 years ago. AC01 harbors 11 types of intact LTR retrotransposons with copy numbers present in both ACs and CCs. The earliest estimated time of genetic exchange between AC01 and CCs is 1 million years ago (Type 3). By analogy, the approximate integration times of AC02 to AC15 range from 0.33 to a maximum of 1.69 million years ago, with 5 ACs inestimable due to insufficient intact LTRs.

## Discussion

Accessory chromosomes (ACs) refer to additional chromosomes that are not necessary for an organism's normal growth and reproduction. ACs have been observed across a wide range of species, including plants, animals, and fungi[8]. In fungi, these chromosomes were first identified in in the pathogenic fungus *Nectria haematococca* during the 1990s[33]. With the advancement of genomics, an increasing number of fungal genomes have been found to contain accessory chromosomes. As of now, 52 fungal species are listed in the B-Chrom database as having accessory chromosomes[34,35], and most of these fungi belong to the phytopathogenic ascomycetes[7,15,17,36]. In the previous study, we discovered that *T. fuciformis* strain Tr01 carries three accessory chromosomes, all of which are mini-chromosomes less than 1 Mb in size[14]. Here, we sequenced 16 complete strains of *T. fuciformis* and assembled 27 haploid genome sets, which include 108 accessory chromosomes, forming 15 groups of homologous chromosomes. The accessory chromosomes exhibit significant diversity in size, composition, and distribution. *T. fuciformis*, a member of the Basidiomycota, spends most of its life cycle in the dikaryotic mycelium stage and can form large fruiting bodies. It is also a dimorphic fungus, existing in both mycelial and yeast forms. This research is the first to reveal the diversity of accessory chromosomes in Basidiomycota. Future in-depth analyses will deepen our understanding of these chromosomes, focusing on their stability and interactions in heterokaryotic states, as well as the potential roles of accessory chromosomes in dimorphic transitions and fruiting body morphogenesis.

Many fungal plant pathogens with ACs exhibit broad host ranges, with the presence/absence of ACs attributed to fitness trade-offs across different hosts. For instance, in *Sinorhizobium rhizophilae*, an AC in strain 16w supports parasitic lifestyles, while its deletion in mutant d16w shifts the interaction to mutualism, enhancing host growth[37]. Similarly, AC-borne genes in *Colletotrichum* species are critical for pathogenicity-knockout of a specific AC gene in *C. asianum* FJ11-1 reduces virulence[38]. ACs also modulate insect pathogenicity: in *Metarhizium robertsii*, horizontal AC transfer between strains enhances competitive advantage and virulence against Argentine ants during co-infection, while ACs in *Beauveria bassiana* contribute to insect host adaptation[11,39].

In our study, 108 ACs across 27 *T. fuciformis* haploid genomes clustered into three distinct groups, which perfectly aligned with the three phylogenetic branches identified via core-gene pangenome analysis. Crucially, association experiments demonstrated that *T. fuciformis* strains only successfully form symbiotic associations with *A. stygium* isolates from the same cluster, with no cross-cluster compatibility-directly confirming symbiotic specificity. Notably, strains compatible with the same *A. stygium* isolate share highly similar AC compositions, and three lineage-specific ACs (AC01, AC12, AC14, the largest in each branch) are exclusively present in strains that form functional symbioses with their cognate *A. stygium* cluster. These observations strongly support the hypothesis that *T. fuciformis* ACs mediate symbiotic specificity with *A. stygium*.

We propose a potential mechanism: AC-borne genes may encode proteins involved in interspecific signaling or metabolic complementation that enable compatible association with *A. stygium*. For *T. fuciformis*, this specificity ensures access to essential nutrients provided by *A. stygium*, while for *A. stygium*, the association may

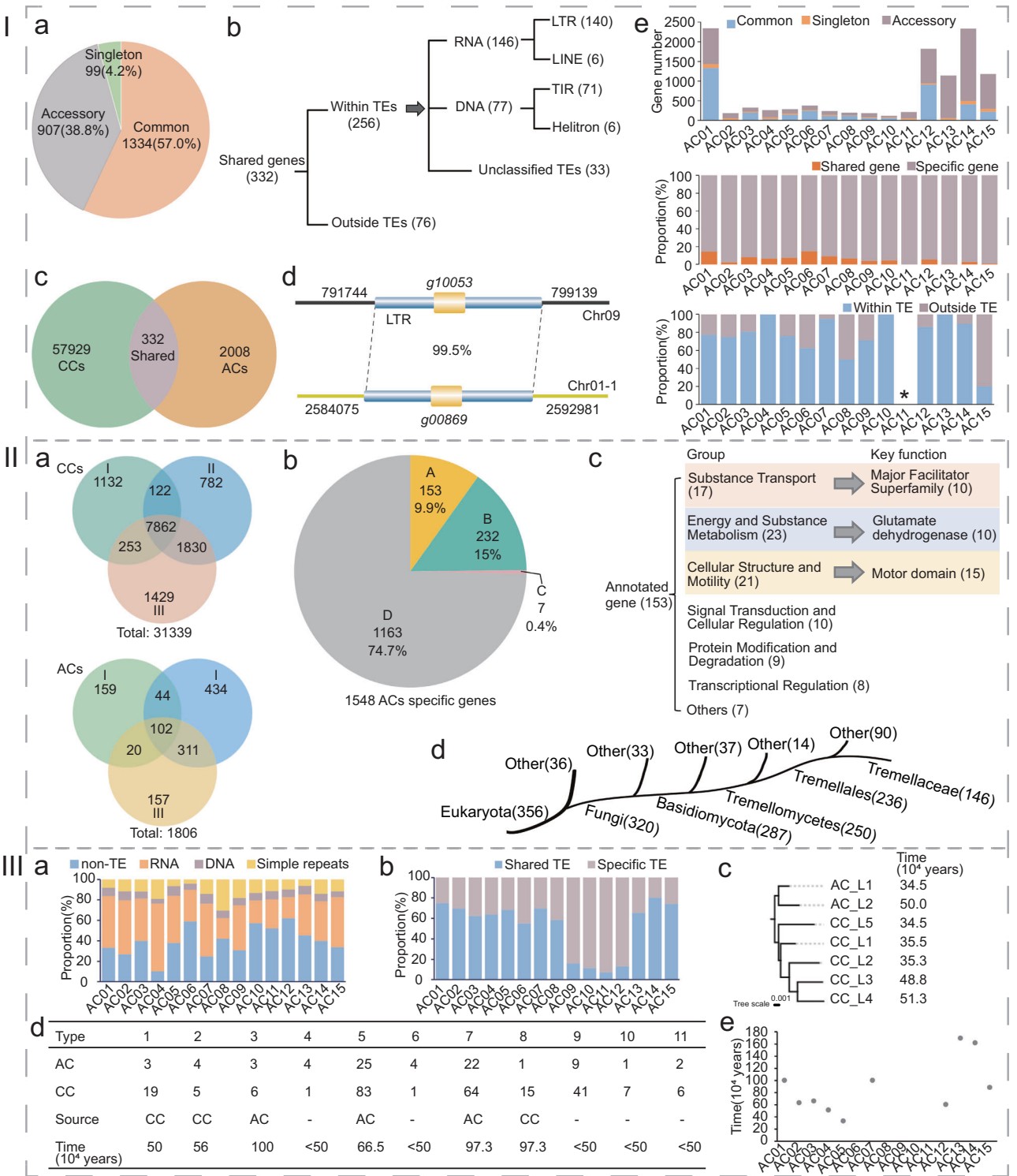

**Fig. 6 | Characterization of accessory chromosome genes and transposable elements in *T. fuciformis*. I** Features of shared genes in accessory chromosomes. **I-a** Pi chart showing the proportions of common genes (shared by all 6 haploid genomes), accessory genes, and singleton genes (unique to a single haploid genome) in the AC01 accessory chromosome. **I-b** Venn diagram showing the number of shared genes between core chromosomes (CCs) and AC01 across six haploid genomes. **I-c** Genomic locations of shared genes. **I-d** Sequence alignment of an LTR transposon harboring a shared gene between core and accessory chromosomes. **I-e** Distribution of common, accessory, and singleton genes in AC01-AC15 (top), shared/unique genes with core chromosomes (middle), and genomic locations of shared genes (bottom). *, ACs with no shared genes (with core chromosomes). **II** Features of AC specific genes. **II-a** Venn diagrams of homologous gene overlaps between AC and CC pan-gene sets across Clusters I-III. **II-b** Distribution of AC-specific genes: annotated (A), functionally unannotated homologous (B), lacked homologous matches and encoded proteins bearing signal peptides (C), and unknown genes (D). **II-c** Enriched biological processes and top abundant functions of annotated genes. **II-d** Taxonomic distribution of homologous AC genes. **III** Features of AC TEs. **III-a** Proportions of repetitive sequence types (non-TE, RNA transposons, DNA transposons, simple repeats) in AC01-AC15. **III-b** Shared TE distribution between each AC and core chromosomes. **III-c** Estimated insertion times of shared intact Type I LTR retrotransposons (with divergent terminal LTRs) in AC01. **III-d** Origins and earliest exchange times of shared intact Type I LTR retrotransposons between AC01 and core chromosomes. **III-e** Earliest divergence times between each AC (AC01-AC15) and core chromosomes. Source data are provided as a Source Data file.

facilitate niche colonization or resource acquisition. To validate this, future experiments should include: (1) targeted knockout of AC01/AC12/AC14 to assess impacts on symbiotic compatibility; (2) transcriptomic profiling of AC-borne genes during *T. fuciformis-A. stygium* interaction; (3) identification of AC-encoded proteins that interact with *A. stygium* cell surface or secreted factors. Such studies will clarify the molecular basis of symbiotic specificity and its implications for both partners' ecological fitness.

Since fungal ACs are not crucial for survival, they often experience processes of deletion and duplication. For example, under in vitro growth conditions, the rate of the spontaneous loss of the accessory chromosome (AC14) in *F. oxysporum* f. sp. *lycopersici* 4287 was about 1 in 35,000 spores. In a study involving the propagation of *Zymoseptoria tritici* isolate IPO323 in liquid culture at 18 °C, after ~80 cell divisions per cell, 38 out of 576 tested strains (~7%) were found to lack one accessory chromosome. Here, we present a more extreme case occurring in TF2206. The TF2206 genome contains three ACs: Chr09, Chr10, and Chr11. We conducted high-throughput sequencing to analyze the sequencing depth of each chromosome in 30 colonies, and the results showed variations in the copy number of ACs across all colonies. Specifically, one colony exhibited a copy number variation in one AC, six colonies showed variations in two ACs, and the remaining colonies displayed copy number variations in all ACs. In these copy number variations, the majority of cases involve an aneuploid decrease in copy number. However, during the process of hyphae transforming into spores, there is an aneuploid increase in the copy number of the Chr10 chromosome in MY2, MY3, and MY6, as well as the Chr11 chromosome in MY8 and MY9. Similar to *Z. tritici* IPO323, this may happen due to improper allocation of accessory chromosomes during mitosis. Considering that TF2206 is a heterokaryon, some cells contain multiple copies of accessory chromosomes, some cells have one accessory chromosome per set, some cells contain only one accessory chromosome, and some cells do not contain the corresponding accessory chromosome at all. When cells with multiple copies of accessory chromosomes are in the majority, the copy number of these chromosomes increases; conversely, the copy number decreases when they are in the minority. This improper allocation of accessory chromosomes during mitosis may further facilitate frequent exchanges between homologous accessory chromosomes in the heterokaryotic strain, resulting in high or even identical sequence similarity among them. Besides the accessory chromosomes, we also found that the core chromosomes of TF2206 frequently exhibit copy number variations. In the Y-Y scenario, the copy number of Chr07 in all yeast isolates doubled, except for the Chr05 of the YY3 colony, which increased by 0.5 times. In the M-M scenario, the copy number of Chr06 in all yeast colonies decreased to 0.7 times. It is suggesting that the copy number of these chromosomes in the original colony has already varied. During the Y-Y transition, the YY3 colony experienced copy number variations in Chr01-2 and Chr05, accounting for 10% of the total colonies. In the M-M transition, no core chromosome copy number variations were observed in any colonies. However, in the M-Y scenario, the copy numbers of Chr02-2 in 8 colonies, Chr05 in MY6, and Chr08 in 3 spores all increased to varying degrees, accounting for 80% of the total colonies. The results indicate that the transition from mycelium to yeasts tends to lead to core chromosome copy number variations. In the three scenarios, the core chromosomes with copy number variations differ, and sequencing analysis of mixed colonies also shows significant fluctuations in the sequencing depth of each core chromosome, but the trend is consistent, suggesting that each core chromosome may undergo copy number variation.

Two non-mutually exclusive models have been proposed to address the origin of accessory chromosomes. One model suggests that they may originate from essential core chromosomes and gradually differentiate or degenerate over time[19], while the other proposes that they could be acquired through horizontal chromosome transfer[4,22,40]. B chromosomes in animals and plants are generally believed to primarily originate from their A chromosomes. This belief is largely supported by their mosaic-like composition, with sequences derived from essential A chromosomes and organelles[41,42]. In some fungi, there is evidence that entire chromosomes are being horizontally transferred between different lineages or even species[26], and this phenomenon has been experimentally confirmed[11,43,44]. The genome data of 16 *T. fuciformis* isolates have provided insights into the possible origin of their accessory chromosomes. The ANI between heterokaryotic genomes of the same strain and between different strains is significantly higher for accessory chromosomes than for core chromosomes, indicating that accessory chromosomes are not inherited from a common ancestor like core chromosomes. When comparing the genes of ACs to those of CCs in each AC type, it was found that less than one-fifth of the AC genes have homologous sequences in the CCs. Among these homologous genes, the majority are transposons. This suggests that ACs may not originate from CCs, and these homologous genes are likely exchanged between CCs and ACs through transposition. We aligned the 1548 AC-specific genes from all strains against the NCBI nr and three functional domain databases and found that over three-quarters of these genes failed to match any homologous sequences or conserved functional domains, suggesting the ACs of *T. fuciformis* may originate from unexplored or understudied species. In the ACs of the three branches of *T. fuciformis*, there are 102 genes that have homologous genes in each branch. Among the 392 AC-specific genes matched on the databases, about two-thirds have the best matches belonging to strains of the order Tremellales. This suggests that the ACs may have existed even before the formation of the *T. fuciformis* species and, after long periods of independent evolution, still retain some homologous genes. Despite the high ANI values of homologous ACs, synteny analysis reveals that they have undergone rapid structural variations during evolution, such as inversions, terminal chromosome loss, insertions-deletions, and chromosome fusion-splitting. This has resulted in the proportion of shared AC genes among the three branches being much lower than that of CCs. In summary, we hypothesize that unlike CCs, the ACs of *T. fuciformis* did not originate concomitantly with the species. Instead, they were acquired through horizontal gene transfer from an unexplored species prior to the species formation, and as non-essential elements, underwent rapid structural variations post-acquisition.

## Methods

### *Tremella fuciformis* strains
We analyzed a diverse set of 16 *T. fuciformis* strains (Supplementary Table 1). Fourteen of these strains were isolated from various regions in China, specifically from the provinces of Fujian, Guangdong, Sichuan, and Yunnan. Tr21 and Tr01 are the primary cultivated varieties of *T. fuciformis*, accounting for over 99% of the total cultivation. Among these strains, 11 exhibit both mycelial and yeast forms, while 5 exhibit only the yeast form. In this study, 15 *T. fuciformis* strains were newly sequenced, and we included one publicly available complete genome sequence (Tr01, which includes the complete haploid A and haploid B genome sequences)[14]. All these strains have been preserved at the Mushroom Germplasm Resource Preservation and Management Center of Fujian Province.

### Sample collection and Genome sequencing
We transferred the mycelial-form strains to PDA plates with a moist surface and incubated them at 25 °C for 3 days. During this period, some cells around the mycelia began to form into yeast forms, which were then transferred to form yeast-form strains. All yeast-form strains were inoculated into PDB medium and cultured with shaking at 150 rpm at 25 °C for 3 days. Spores were collected during the mid to late logarithmic growth phase. The yeasts were harvested by centrifugation at 10,000 g for 10 minutes and washed twice with sterile

double-distilled water. The collected samples were used for PacBio CCS sequencing using the Sequel IIe platform, aiming to obtain ~10 Gb of high-quality HiFi reads for whole-genome assembly. DNA extraction, quantification and evaluation, library construction, and sequencing were all conducted by Frasergen Bioinformatics Co., Ltd. (Wuhan, China).

Sixteen *T. fuciformis* strains were divided into three clusters, with the Tr01 strain located in Cluster III, for which a complete genome sequence has been obtained. NDB and TF2206, randomly selected as representatives of the Cluster I and Cluster II, were first subjected to Hi-C-assisted genome assembly to obtain complete and accurately phased genomes. They then served as reference genomes together with Tr01 to assist in the genome assembly of other strains within their respective clusters. The samples used for Hi-C sequencing were collected in the same way as those used for PacBio CCS sequencing. All steps of the Hi-C sequencing process, including cell fixation, chromatin digestion, fragment ligation, cross-link reversal, DNA purification, and high-throughput sequencing, were carried out by Frasergen Bioinformatics Co., Ltd. (Wuhan, China).

Using the TF2206 strain as an example, we analyzed the chromosomal copy number variations in *T. fuciformis* during the proliferation of yeast-like cells, the proliferation of mycelial cells, and the transition from mycelial to yeast-like cells. In the yeast-like cell proliferation experiment, we selected a single yeast-like colony of TF2206, diluted it in sterile water, plated it, and after cultivation, randomly picked 10 single yeast-like colonies. In the mycelial cell proliferation experiment, we selected a single mycelial colony of TF2206, picked 10 mycelial blocks, and transferred them to PDA plates to form 10 new mycelial colonies. In the experiment of transitioning mycelial cells to yeast-like cells, we selected a single mycelial colony of TF2206, picked 10 mycelial blocks, and transferred them to moistened PDA plates. During cultivation, some cells around the mycelia began to form yeast-like structures, and single yeast-like colonies were obtained through streak purification. The cultivation temperature for mycelial cells, yeast-like cells, and the transition from mycelial to yeast-like cells was consistently maintained at 25 °C. These colonies were collected and sent to Annoroad Gene Technology Co., Ltd. (Beijing, China) for Illumina high-throughput sequencing. The processes of DNA extraction, quality control, library construction, and paired-end sequencing on the Illumina NovaSeq 6000 platform (2 × 150 bp) were all completed by Annoroad Gene Technology Co., Ltd. (Beijing, China).

### Genome assembly

For the TF2206 and NDB strains, we used the Hi-C mode of the HiFiasm (v0.15.3-r339)[45] to assemble the sequenced HiFi reads with default parameters and ultimately obtaining two sets of contigs, hap1 and hap2, each of which was further assembled individually. We manually examine whether the 5′ end of each contig contains a telomeric sequence with the repeat unit TAACCCCC, and whether the 3′ end contains a telomeric sequence with the repeat unit TTAGGGGG. These telomeric sequences help determine if the contigs form complete chromosomes and allow for a preliminary estimation of chromosome numbers. Contigs that do not form complete chromosomes will be used for further analysis. The mitochondrial genome sequence of *T. fuciformis* TF01 (GenBank accession: NC_036422.1) was downloaded from the NCBI website. Using local BLAST, it was compared with the remaining contigs to identify those related to the mitochondria. Since the mitochondrial genome of *T. fuciformis* is generally less than 50 Kb[46], a single contig can cover the entire mitochondrial genome sequence. The complete mitochondrial genome is obtained by removing the repeat sequence at one end. These contigs were then manually assembled into circular mitochondrial DNA. Each haploid genome contains an rDNA region, composed of tandem repeats of ~10Kb rDNA units[47], located in the middle or at the end of a chromosome. The ITS sequence of *T. fuciformis* CBS 6970 (GenBank accession:

NR_155936.1) was retrieved from nt database in NCBI website and compared with the remaining contigs using local BLAST to identify contigs carrying the rDNA region and exclude contigs that fall within the rDNA region. When two contigs carrying the rDNA region are found, they are considered to be connected, forming a new scaffold. If only one contig with the rDNA region is identified, the presence of rDNA at the chromosome end is determined by searching for HiFi reads that contain both rDNA repeat units and telomere repeat sequences. The number of tandem repeats of the rDNA unit is calculated as the ratio of the sequencing depth of the rDNA unit to that of the single-copy sequence. Hi-C reads were utilized to identify connections between contigs within a scaffold containing gaps, using the programs HiC-Pro (v2.10.0)[47] and HiCPlotter (v0.7.3)[48]. After completing the scaffolding process, manual correction was employed to address the existing gaps. This involved trimming the ends of adjacent contigs and performing overlapping alignments to achieve seamless connections between the contigs. The phasing of the two sets of chromosomes was performed using program NuclearPhaser. To rule out errors caused by genome assembly, the observed chromosomal rearrangements were validated using orthogonal approaches, including long HiFi read mapping, long nanopore read mapping, Hi-C interaction heatmaps, and/or long-range PCR assays.

For the remaining 13 strains, we used the HiFiasm in non-Hi-C mode to assemble their sequenced HiFi reads into contigs using the default parameters. The determination of complete chromosomes, preliminary estimation of chromosome numbers, assembly of the mitochondrial genome, and rDNA-related analyses are all consistent with the analysis procedures of TF2206 and NDB. For the strains of Cluster I, II, and III, the complete genome sequences of TF2206, NDB, and Tr01 were selected as references, respectively. Through synteny analysis, the remaining contigs were assembled to the chromosomal level. Subsequently, sequence similarity analysis was employed to classify all chromosomes of heterokaryotic strains into two genomic sets.

### Identification of accessory chromosomes

Accessory chromosomes are identified in this study by comparing 27 haploid genomes. Chromosomes present in all strains are classified as essential, while the others are categorized as accessory. The comparison of gene synteny between Tr01 hapA and the other 26 haploid genomes was conducted using the MCScanX (https://github.com/wyp1125/MCScanX)[49]. The annotation GFF files of each genome were first converted into standardized BED format using the jcvi.formats.gff subroutine. Subsequently, local BLAST databases were constructed and aligned to identify homologous genes across genomes. Based on the alignment results, the MCScanX algorithm was employed to screen high-similarity anchor genes while precisely determining their relative genomic positions, thereby detecting conserved syntenic blocks. To visualize genome-wide collinearity patterns, conserved syntenic gene clusters with evolutionarily stable arrangements were extracted using the group_collinear_genes.pl script from the MCScanX pipeline, followed by the generation of high-resolution syntenic dot plots via the dot_plotter.java program.

### Gene annotation and TE prediction

Gene annotation was performed on 27 *T. fuciformis* genome assemblies using FUNANNOTATE (v1.8.16)[50]. This tool integrated homologous protein sequences and full-length transcript evidence from the reference genome Tr01 Parameters were set to specify the species as *T. fuciformis*, with the closely related species *Cryptococcus neoformans* JEC21 selected as the training set for Augustus.

To characterize the gene composition of accessory chromosomes, we first aligned AC-encoded genes against a local non-redundant (nr) database using DIAMOND (v2.1.8)[51] with an E-value threshold of 0.1, and analyzed the taxonomic relationships of the

top-scoring hits. Additionally, InterProScan (v.5.67-99.0)[52,53] was employed to query these genes against three databases (Pfam 36.0, SMART 9.0, and MobiDBLite 2.0), facilitating the identification of conserved protein domains, functional motifs, signal peptides, transmembrane domains, low-complexity regions, and intrinsically disordered regions. Complementary predictions were performed using DeepTMHMM (v.1.0)[54] and SignalP 6.0[55] to determine the transmembrane domains, signal peptides, and secretion types of AC-encoded proteins. Collectively, these analyses yielded a comprehensive characterization of the genes within accessory chromosomes.

Transposable elements in each of the 27 haploid genomes were annotated using a combination of tools: LTRharvest (GenomeTools) 1.6.5[56], LTRdigest[57], RepeatModeler (http://www.repeatmasker.org), and EDTA v2.2.2[58]. All annotations were merged to construct a *T. fuciformis*-specific TE database. This species-specific database was then used in conjunction with RepeatMasker (v4.0)[59] to perform genome-wide annotation of repetitive sequences across the 27 haploid genomes.

Starship giant transposons in 27 haploid genomes were identified using Starfish (v.1.0.0)[60] with multiple modules. Candidate tyrosine recombinase (YR) genes (potential captain genes) were retrieved and filtered via the annotate module, leveraging the built-in YRsuperfams hidden Markov model database (YRsuperfams.p1-512.hmm) and reference protein database (YRsuperfamRefs.faa). The consolidate module integrated original genome annotations with YR candidate annotations. The sketch module scanned YR genes and their 10 Kb flanking regions to preliminarily define the boundaries and gene composition of potential Starship elements. Using YR genes as anchors, the insert module performed similarity searches against a pre-constructed BLAST database (parameters: ≥80% nucleotide identity, ≥750 bp HSP length) to determine insertion start/end positions. Candidate Starship regions were confirmed by manual inspection.

### Pan-genome analysis and phylogenetic tree construction

To identify orthologous groups (OGs), we employed OrthoFinder (v2.5.5)[61] to analyze the predicted proteins from the 27 haploid genomes. We used MAFFT (v7.525)[62] to perform homology alignment of the protein sequences from all genomes, with parameters set to default mode. Subsequently, OrthoFinder was used to generate multiple sequence alignment files, which were then integrated into a superalignment matrix using the -M msa parameter. OGs present in all genomes were classified as core genes, those found only in some genomes were considered accessory genes, and those existing in a single genome without clear orthologs were categorized as singletons. The phylogenetic relationship among 27 *T. fuciformis* haploid genomes was inferred using all core genes. Based on the alignment results, a maximum likelihood (ML) phylogenetic tree was constructed with FastTree (v2.1.11)[63], using the parameter -m set to MFP and performing 1000 bootstrap resampling. Visualization was done with the web-based ChiPlot (v2.6.1, https://www.chiplot.online/)[64], where the tree layout was manually optimized and key evolutionary branches were annotated.

The ITS sequence of *T. fuciformis* strain CBS 6970 was used as the query to align with the PacBio sequencing reads databases of 16 strains via the local BLAST program. Fragments with ≥5 identical sequences in matches were screened, identified as ITS sequences of the corresponding species, and truncated. These truncated ITS sequences served as new queries for searching the online UNITE database (https://unite.ut.ee/schedule_analysis.php). The most similar homologous sequences were downloaded for subsequent phylogenetic tree construction. Sequence analysis was performed using MEGA 11 (v11.0.13)[65]: All ITS sequences were aligned via the built-in ClustalW tool with parameters set as gap opening penalty = 6.66 and gap extension penalty = 15. A phylogenetic tree was constructed based on the ML method, with the optimal tree structure searched via the Close

Neighbor Interchange (CNI) strategy; its reliability was evaluated using 1000 replicate bootstrap tests. Finally, the online tool ChiPlot was used for phylogenetic tree visualization. The same alignment, tree-building, and visualization methods were applied to construct the phylogenetic tree of mitochondrial genomes using the complete genome sequences of the 16 strains.

### Average nucleotide identity analyses

We utilized the Average Nucleotide Identity (ANI) method to analyze the differences between phylogenetic relationships of homologous core chromosomes and those of homologous accessory chromosomes in the *T. fuciformis* haploid genomes. The ANI values and coverage were calculated using the online OrthoANIu (https://www.ezbiocloud.net/tools/ani)[66], with chromosome sequences as input. A single-cycle comparison approach was employed, where each homologous chromosome was compared with all other homologous chromosomes, to assess the phylogenetic relationships within the same set of homologous chromosomes.

### Chromosomal copy number variation analyses

Illumina sequencing reads were aligned to the TF2206 hapA genome using BWA-MEM (v0.7.18)[67] with default settings for accurate mapping. Post-alignment, GATK (v4.0.11.0)[68] was employed for several processing steps: creating a sequence dictionary, rearranging SAM files, sorting BAM files, and marking duplicates to avoid biases in variant calling. Samtools (v1.15.1)[69] converted SAM files to the more compact BAM format and calculated the sequencing depth of the final BAM file, offering insights into genome coverage. Chromosome copy number variations were identified by dividing each chromosome's sequencing depth by the genome's average sequencing depth.

### Chromosomal structural variation analyses

To analyze chromosomal structural variations, genomic data in GenBank format were used as input files. Genome sequences were aligned using nucmer from the MUMmer (v4.0.0)[70], applying a length threshold of 2000 to exclude shorter alignments and focus on significant ones. Syntenic relationships and structural variations were visualized using PyGenomeViz (v1.5.0) (https://github.com/moshi4/pyGenomeViz), which effectively illustrates complex genomic rearrangements. The analysis focused on structural variations such as inversions, terminal chromosome losses, insertions-deletions, and chromosome fusion-splitting events.

### Analysis of accessory chromosome origins

To investigate the origins of ACs, we addressed three key questions through the following analyses.

For distinguishing between core chromosome derivation and external acquisition, we identified genes with homology to core chromosome genes via sequence alignment for each AC (AC01 to AC15). Using AC01 as a representative, we analyzed all genes in both this accessory chromosome and the core chromosomes of its corresponding strains, clustered them into orthologous groups (OGs), and calculated the proportion of genes with homology. We then mapped these homologous genes on the accessory chromosomes to determine whether they were embedded in transposable element sequences and transferred between core and accessory chromosomes via TE-specific loci[71]. Based on the results of these two analyses, we inferred whether the AC01 originated from core chromosomes. This analytical framework was similarly applied to AC02 to AC15.

For identifying potential donor species, we analyzed the all AC-specific genes by conducting homology searches against public databases to classify genes as having known or unknown functions; using InterProScan, DeepTMHMM, and SignalP 6.0 to annotate conserved domains, functional motifs, signal peptides, transmembrane domains, low-complexity regions, and intrinsically disordered regions; and

evaluating the taxonomic distribution of top hits from homology searches, with specific attention to sequence similarity to species within Tremellales and other related taxa, complemented by functional enrichment analysis on annotated genes to characterize over-represented biological processes.

For estimating integration timing into *T. fuciformis*, we analyzed repetitive sequences in AC01 to AC15, focusing on TEs and quantifying the proportion of TEs shared with core chromosomes versus AC-specific TEs; for AC01, we characterized intact LTR retrotransposons shared with core chromosomes, inferred their origins, and estimated insertion times using sequence divergence-based method implemented in MEGA11 (Jukes-Cantor model)-specifically, employing a neutral mutation rate ($r = 2 \times 10^{-9}$ substitutions per site per year) derived from *Cryptococcus neoformans*[72], with this approach replicated for AC02 to AC15 to approximate their integration timing into the *T. fuciformis* genome.

### Association experiments between *T. fuciformis* and *A. stygium*

Among the 16 *T. fuciformis* strains examined, 11 were able to form compatible associations with their corresponding *A. stygium* strain. Based on the phylogenetic placement of *T. fuciformis* (Clusters I, II and III), the 11 corresponding *A. stygium* strains were classified into three groups: AF1, AF2 and AF3. Each *A. stygium* was inoculated onto a sawdust-based medium and incubated at 25 °C for 7 days until full mycelial colonization was achieved. The colonized sawdust blocks were then transferred to fresh Petri dishes and gently compressed. The cultures were incubated for an additional 3 days to allow for mycelial recovery and adaptation. A single-cycle inoculation method was employed, in which all 11 *T. fuciformis* strains were individually inoculated onto the same *A. stygium*-colonized sawdust substrate within each dish for co-cultivation. The co-cultures were incubated at 25 °C for 20 days. Successful associations were identified by the appearance of dense, white, cotton-like mycelial tufts at the margins of the *T. fuciformis* inoculation sites, indicating a compatible interaction between *T. fuciformis* and *A. stygium*.

### Reporting summary

Further information on research design is available in the Nature Portfolio Reporting Summary linked to this article.

## Data availability

The sequencing reads of 15 *Tremella fuciformis* strains generated in this study have been deposited in NCBI BioProject PRJNA1247727. Genomic sequences (mitochondrial and nuclear genomes) of these strains have been deposited at the National Genomics Data Center (NGDC) under accession PRJCA052213. The mitochondrial genome sequence of Tremella fuciformis TF01 used in this study is available in the GenBank database under accession code NC_036422.1 (https://www.ncbi.nlm.nih.gov/nuccore/NC_036422.1). The ITS sequence of Tremella fuciformis CBS 6970 used in this study is available in the GenBank database under accession code NR_155936.1 (https://www.ncbi.nlm.nih.gov/nuccore/NR_155936.1). Source data are provided with this paper.

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

## Acknowledgements

We are deeply grateful to the experimental team of our research group for their continued support and valuable insights throughout this study. We especially thank Prof. Ray Ming, Prof. Baogui Xie, and Prof. Xiaoping Wu for their valuable contributions to study design and conceptual development. We sincerely acknowledge Zhiwen Lv (State Key Laboratory of Microbial Diversity and Innovative Utilization) for providing insightful suggestions on data analysis. We also thank Xingcai An (Yunnan University) for kindly providing two *Tremella fuciformis* strains collected from Yunnan. This work was supported by the National Natural Science Foundation of China (32472810, Y.D.) and the Tongjiang County Edible Fungi Scientific Research Cooperation Project (N5119212023000037, Y.D.), and the Major Science and Technology Project of Fujian Province (2022NZ029015, X.W.).

## Author contributions

Y.D., R.M., B.X., and X.W. conceived and designed the study. J.Z., Y.D., X.A., and H.H. collected and isolated microbial strains. J.Z., Q.T., and F.L. performed experiments and data collection. J.Z., Y.D., T.Z., and Z.L. conducted genome assembly and polishing. J.Z., Y.D., Q.T., H.C., J.Y., X.L., H.X., and F.Z. analyzed and interpreted the data. J.Z. and Y.D. drafted the manuscript and prepared the figures. All authors critically reviewed, edited, and approved the final version of the manuscript.

## Competing interests

The authors declare no competing interests.
