## [Transparent Peer Review file · Nature Communications]

Polymorphism and evolutionary origins of accessory chromosomes in the basidiomycete *Tremella fuciformis*

Corresponding Author: Professor You Deng

Version 0:

Reviewer comments:

Reviewer #1

(Remarks to the Author)

The manuscript "Exploring the polymorphism and origins of accessory chromosomes in *Tremella fuciformis*: insights into genome divergence and structural variation" by Jinxiang Zhang and co-authors presents a fascinating story on the evolution of accessory chromosomes (ACs) in *Tremella* basidiomycetes. Authors annotated 16 strains (27 haploid genomes) to assess the trends of ACs abundance among individual genomes and compare their similarity to the core chromosomes. The manuscript presents several interesting and novel findings. It is worth mentioning that accessory chromosomes in fungi are known from Pezizomycotina and the report extends the taxonomic range of their occurrence to Basidiomycota/Agaricomycotina. The collection of ACs spans elements of diverse size (from 200kb to chromosome size objects), different degree of identity to the core chromosomes and conservation among isolates.

As expected the ACs are enriched in transposons and have a lower gene density than core chromosomes. However there are signs of AC coevolution with the core genome, namely closely related isolates harbored the same or almost the same set of ACs. Authors report a very small set of common genes in the pangenome (24.3%) and huge differences in genome size between the genomes 32.4 v 23.8Mb.

The study is well written, robust and straightforward. The main shortcoming is that the manuscript ends briefly, just before I expected to find an analysis of the gene and transposon content in accessory chromosomes and further interpretation of how these chromosomes originated and evolved.

I would like to express only a few concerns /questions/invitations to extend the study:

Please provide a systematic analysis of the gene content in terms of gene function and properties of encoded proteins. If ACs follow the trends of the two speed genome it is expected to see repeat-rich, low complexity rich, disordered, effector-like and transposon-related sequences in the ACs more often than in the core chromosomes.

Which evolutionary scenario leads to the lower ANI observed in the ACs?

Starships are shaping fungal genes and contribute to the moves of massive parts of the genomes. Have you found Starships in the *Tremella* genomes? If yes, how do they relate to the ACs? If you have not searched for them so far you might use the dedicated software starfish available on github.

Is the pattern of intronization of genes similar between ACs and core chromosomes? Does the gene pool differ in terms of length, intronization, gene structure and function?

Please comment on the ACs gene and transposon content across the taxa. Are all TEs equally likely to be localized in ACs and core chromosomes?

Fig.4 lines 292-293 describe the location of the chromosome fusion. However there is no information about what exactly is there, is it a long transposon, repeat region or a genic region?

The description of TE content in lines 352-353 does not explain the diversity of TEs in the ACs compared to the core chromosomes. Please report which families of TEs are shared and which are unique to each of the chromosome types.

Furthermore, please describe TE family proliferation and if possible date the events for LCR retrotransposon bursts.

When delimiting fungal species it is recommended to use UNITE as a source of curated ITS references.

The description of gene content in lines 364-369 is not exhaustive. No hits in nr NCBI database does not necessarily show they are orphan genes. One possible path to describe these genes is to cluster the gene pool of the ACs to see how many are shared and how many are specific to an individual AC type. Another avenue is to map the genes on domain databases and assign them to protein families and folds. Having such annotation allows for a better prediction of protein function.

Complementary to domain analysis it is recommended to use non-domain protein features such as secretion signal, cellular localization, transmembrane helices, coiled coils, disordered and low complexity regions to describe the encoded protein set

in ACs relative to core chromosomes. I would suggest to perform all three annotation procedures to describe in detail the content of ACs and compare them.

Reviewer #2

(Remarks to the Author)

Accessory chromosomes, also known as dispensable, lineage-specific, supernumerary or B-chromosomes, are thought to be non-essential for normal growth and/or development. These chromosomes occur in several eukaryotes, but their evolution and function has been extensively studied in ascomycete fungi, and especially in plant pathogens. Basidiomycetes are another major fungal lineage, comprising most of the mushroom forming species. In contrast to ascomycetes, however, the presence, function, and evolution of accessory chromosomes are not well documented.

The manuscript 'Exploring the polymorphism and origins of accessory chromosomes in *Tremella fuciformis*: insights into genome divergence and structural variation' by Zhang and colleagues reports on the genome sequencing and assembly of haplotype-resolved genome sequences of 16 strains of the basidiomycete fungus *Tremella fuciformis*. Using this resource, the manuscript studies their chromosomal makeup, identifying abundant accessory chromosomes. It provides an in-depth description of the unique genomic characteristic of accessory chromosomes and observed that most accessory chromosomes lacking identifiable homologs in other species. Based on these observations, the manuscript discusses potential origins of these accessory chromosomes as well as the implications for *T. fuciformis*'s biology.

The observation of abundant accessory chromosomes in a basidiomycete fungus is very interesting, broadening the common idea of their occurrence and importance solely in ascomycetes. Moreover, the generated genomic resources will be relevant for the fungal community. However, I have several concerns about the novelty of the observations, the robustness of the presented data, and the general clarity of the manuscript.

Most importantly, nearly the same group of authors have previously published a manuscript in *Genome Biology* (<https://doi.org/10.1186/s13059-023-03093-7>) reporting on the presence of accessory chromosomes and detailed many of their characteristics. That manuscript had also already reported copy-number variation in core and accessory chromosomes as well as structural variations, alike much of the data presented here. However, the current manuscript is not transparent which observations are rather confirmatory to the previous work and what are truly unique and novel, and how these make a significant contribution to our understanding of accessory chromosomes in basidiomycetes. I do understand that the current submission is more exhaustive, providing data from different strains and haplotypes, but the true novelty is not apparent.

There are several experimental approaches that are not clearly explained, nor their impact is analyzed and/or discussed. Furthermore, some observations need more detailed explanations. For example, the manuscript reports the ITS tree for the 16 strains (L121ff), but it is not clear how the ITS is derived from dikaryotic strains that should have two ITS; also, NJ algorithm is used rather than more sophisticated tree reconstruction approaches. Hi-C data is generated for NDB and TF2206, however it is not explained why these strains were selected. Tr01 was used as a reference to place NDB contigs into chromosomes. Given the known chromosomal rearrangements (Deng et al. 2023), this could potentially introduce error in the assemblies; similarly, the Hi-C enhanced genomes were used to place contigs for other (L553). How was the potential issue addressed or corrected? Observed chromosomal rearrangements could for example be validated (e.g., via orthogonal approaches such as PCRs). It is also not entirely clear how the core chromosomes were identified (L164), as I would expect a working definition would be that these need to be conserved in all strains. It seems that the number differs between eight and ten, while this observation is only explained much later (and had been as far as I can tell reported previously already; Deng et al. 2023); the implication of the transposons at the breakpoints/junctions are not further detailed but could provide insights in the processes driving chromosome variation. For additional comments, see further details below.

The manuscript needs to overall improve in clarity. Just as an example, the abstract is illegible for non-expert readers. It is not explained what accessory chromosomes are in comparison to core chromosomes, what *T. fuciformis* is, or why 16 strains would yield 27 haplotypes (monokaryon vs dikaryons). The abstract mentions 'homology similarity' (L31), but this is not a common concept since homology is qualitative, but similarity is a quantitative trait. Similarly, parts of the introduction are superficial, repetitive (e.g., L59 vs L66 or L73 saying the same as few lines above), or plain wrong. For example, L48 implies that all fungi have unique characteristics such as the two-speed genome phenomenon, while there is plenty of evidence that this is not common to all fungi and differ significantly between even closely related fungi. In contrast to the statement of L59f, accessory chromosomes in *Z. tritici* have to my knowledge not been demonstrated to contain virulence factors or secondary metabolite clusters, and their functions remain enigmatic. Accessory genomic regions have been successfully identified using (epi-)genomic characteristics alone (in *Magnaporthe* and *Verticillium*) and thus the statement in L75 is also incorrect.

Throughout the manuscript (e.g., L127), the companion fungus *Annulohyphoxylon stygium* is mentioned several times by their significance to *T. fuciformis* remains unclear. It is suggested that there might be a link to the presence of a subset of accessory chromosomes (e.g., Fig 2), but the manuscript remains vague with respect to mechanisms and further implications for both partners. Moreover, the material & methods implies that the authors performed mating experiments, however, these seems to be rather association than true mating. The authors need to be much clearer on the presence, impact, and implication of the companion fungus to the genomics and biology of *T. fuciformis*.

Additional comments and suggestions:

Title: for the general audience, spell out what Tremella is. Moreover, the content before and after the colon seems to be redundant.

L61: Does this imply that there are basidiomycetes with accessory chromosomes? If so, this needs to be introduced in much more detail.

L120: What are '...principle cultivated varieties,...'? Please explain.

L124: The sentence is unclear as it suggests that ITS sequences exhibit morphological characteristics. Please rephrase.

L138: '... modern...' not sure what this refers to, could this be an error?

L202: The manuscript should be clearer when it reports on haploid genomes. Does it refer to a monokaryotic strain or to a dikaryotic strain where both haplotypes have been assembled? This distinction could also be clearer in the figures (e.g., the A and B in the strain name)

L229: this pattern could also be due to horizontal transfer of the chromosome rather than loss. Could the authors perform additional analyses to clearly distinguish these scenarios?

L262: What is driving the high similarity between accessory chromosomes? Is this mainly due to transposons or also over the protein-coding (non-TE) genes?

L279: it needs to be clear that the two AC14 refer to the A and B haplotype rather than a copy-number variation of one of the two.

L348: Why was C-1 used as a case study?

L367: Additional information about the genes on accessory chromosomes are lacking. For example, are these expressed, are these predicted to be secreted, etc. Do they have other characteristics that makes them distinct from those on the core genome (e.g., number of introns, length, etc.). Moreover, the absence of any homology raises important questions about their origin that unfortunately remains unclear. I find it surprising that most genes do not even have homologs in other fungi and the explanation that we lack the corresponding species in the database seem unlikely given the ongoing sequencing efforts.

L399: Does this sentence imply that this are plans? If so, please rephrase as this is not entirely clear.

L351: I find it odd to read that transposons are treated similarly to genuine protein-coding genes. The abundance of these in the gene sets suggest that the structure gene annotation was not performed carefully, and transposons should be removed prior to any analyses

L519: '...did not exist from the beginning...' please rephrase

L590: few details on the mitochondrial assemblies could be provided. Do they similarly display three distinct groups. Are phylogenies with mitochondrial markers concurrent to nuclear taxonomy?

Reviewer #3

(Remarks to the Author)

In this work the authors describe sequencing of 27 haplotypes from 16 strains of the basidiomycete *Tremella fuciformis*, and a subsequent pangenomics investigation of core and accessory chromosome statistics and distribution.

Major comments

I think the primary critique involves the relationship between accessory chromosomes and *A. stygium* pairing, which is one of the more interesting biological aspects of this work. It is very interesting that the distribution of ACs correlate with the pairing groups, and this is a valid and potentially novel finding.

However, supplementary table 1 introduces 4 strains that do not associate with *A. stygium*: Wuyi2014, Wuyi2015, Lys2020, and T0053. Firstly, the statements in the introduction (lines 101-103) seem to imply that all *Tremella* species associate with *Annulohypoxylon*, so it might be worthwhile to point out here (and potentially also in lines 127-129 in the results) that not all *Tremella* associate with *Annulohypoxylon*.

But overall, there is very little discussion about how the distinct accessory chromosome patterns relate to these four non-pairing strains, as they compare to their pairing-capable phylogenetic neighbors.

For example, Figure 4 shows structural variations in AC01 (inversion) and AC08 (deletion) between NDBA (a pairing strain) and Wuyi2015 (a non-pairing strain), but there is no discussion or speculation as to whether such differences may influence pairing.

Additionally, and specifically with respect to one of the major conclusions, that the three largest ACs (AC01, AC12, and AC14) correspond to the three distinct AF groups (AF2, AF1, AF3), there are nuances that are not addressed. For example,

AC14 is missing from non-pairing strain T0053 but present in non-pairing LYS2020. Similarly, AC01 is present in both non-pairing strains Wuyi2014 and Wuyi2015.

There is also no discussion of other nuances observed in figure 2 and how they could relate to *A. stygium* pairing capabilities. For example, AC08 is primarily found in AF2, but also found in two AF3 strains LYS2020 and T0053, which are notably non-pairing. Similarly, AC06 and AC08 are absent in non-pairing Wuyi2014 but present in non-pairing Wuyi2015. Finally AC15 doesn't show exact group specificity, since it is present in both AF2 and AF3 (excluding B-1 and C-1).

I think that at minimum, some inclusion of these observations in the discussion is warranted. I think it would be beneficial as well to include additional structural comparisons similar to Figure 4, between other closely related strains that differ in pairing ability. Or even a comparison and discussion of how the LYS2020 and T0053 AC08 chromosomes might differ from the AC08 chromosomes found in any of the AF2 strains might be informative.

Minor comments:

- Figure 1d: I don't see any black circles in either chart
- Figure 2: "Dots indicate the absence" should probably be corrected to "minus signs indicate the absence"
- Chromosomal copy number variation Section:
 - Why was only TF2206 selected? Many other strains exist in M/Y morphotypes
 - Why was only YY2 selected for the normalized depth distribution curve comparison (Figure 5d)?
 - The section states that "Ten single colonies" were selected for sequencing from strain TF2206. However the ten colonies are subsequently themselves referred to as "strains" (e.g. "YY3 strain", "YY1 strain"). To distinguish these from the broader biological strains ("TF2206", "TF104", etc) maybe they should be referred to as "colonies"?
 - Line 315-316: "obtained from subculturing the same colony" -> it was unclear which colony is "the same" colony?
 - In Figure 5: The "M to Y" label makes sense because this is described as coming from a mycelial to yeast transition. But "Y to Y" and "M to M" do not really make sense because these were isolated simply as yeast or mycelia, with no transitions involved.
- Figure 6a
The Venn diagram is just the data from the C-1 row. So, why is CCs 8,653 in both cases, but ACs is 297 in Venn diagram and 318 in table? The value of CCs should be different between the Venn diagram and the table, depending on if 8,653 is the total count or the unique count.

Version 1:

Reviewer comments:

Reviewer #1

(Remarks to the Author)

The revised version of the manuscript "Exploring the polymorphism and origins of accessory chromosomes in basidiomycete *Tremella fuciformis*: insights into genome divergence and structural variation" provides new findings. One of the most exciting results is the description of eight Starships integrated both to core and accessory chromosomes, including one exactly at the junction locus. Authors answered all the reviewers questions, introduced requested changes and performed additional analyses which resulted in a significant improvement of the manuscript.

Minor:

I. 292 TIR/Mutato should be TIR/Mutator

LTR/Gypsy should be replaced along the manuscript with LTR/Ty3 because of the discriminatory character

data availability:

annotated assemblies should be deposited; the BioProject: PRJNA1247727 stores the raw reads

Reviewer #3

(Remarks to the Author)

The revised manuscript by Jinxiang Zhang and co-authors thoroughly addresses the comments I made in my original review. In addition, the manuscript has been carefully and substantially improved with enhanced methods, new and expanded analyses, and refined clarity in the text. I have no further comments to offer.

Core Revisions & Highlights

1. Advanced understanding of *T. fuciformis* accessory chromosome (AC) origins: Expanded the prior simplistic analysis with comprehensive annotations of repetitive sequences/gene composition across 27 haploid genomes and systematic core chromosome (CC) comparisons to address three key questions (origin, donor species, integration timing)-the first dedicated in-depth analysis of AC origins, with evidence for horizontal transfer from understudied taxa-supported by a newly generated Figure 6 and fully rewritten corresponding Materials and Methods/Results sections.

2. Novel insights into Starship giant transposons in *T. fuciformis*: Conducted systematic identification of Starship giant transposons via Starfish software across 27 haploid genomes, revealing 8 elements (3 on core chromosomes, 5 on ACs) with functional implications-including potential mediation of chromosome fusion (SS04) and structural variation patterns that support rapid AC divergence-representing the first analysis of Starships in this species and their association with AC dynamics.

3. Rigorous validation of chromosomal rearrangements: Implemented multiple orthogonal approaches (HiFi/nanopore read mapping, Hi-C heatmaps, multi-strain synteny analysis) to confirm rearrangements, addressing reference bias concerns and enhancing result reliability.

4. Methodological optimization for ITS phylogenetics: Updated tree construction from NJ to Maximum Likelihood (ML) with 1000 bootstrap replicates, resolved heterokaryotic strain ITS ambiguity via PacBio read-specific identification, and utilized the UNITE database as curated ITS references for accurate fungal species delimitation.

5. Enhanced manuscript clarity, accessibility, and rigor: Revised the abstract to clarify key concepts (core/accessory chromosomes, *T. fuciformis* as silver ear fungus, 27 haplotypes' origin) for non-experts, corrected "homology similarity" to standard "sequence similarity," and refined the Introduction by rectifying overgeneralizations (e.g., "two-speed genome" limited to "some fungi"), correcting incorrect claims about *Z. tritici* accessory chromosomes, resolving redundancy, and aligning identification-related statements with established literature.

Point-by-point responses to Reviewers' comments.

Reviewer#1

Q1:

The manuscript “Exploring the polymorphism and origins of accessory chromosomes in *Tremella fuciformis*: insights into genome divergence and structural variation” by Jinxiang Zhang and co-authors presents a fascinating story on the evolution of accessory chromosomes (ACs) in *Tremella* basidiomycetes. Authors annotated 16 strains (27 haploid genomes) to assess the trends of ACs abundance among individual genomes and compare their similarity to the core chromosomes.

The manuscript presents several interesting and novel findings. It is worth mentioning that accessory chromosomes in fungi are known from Pezizomycotina and the report extends the taxonomic range of their occurrence to Basidiomycota/Agaricomycotina. The collection of ACs spans elements of diverse size (from 200 Kb to chromosome size objects), different degree of identity to the core chromosomes (CCs) and conservation among isolates.

As expected the ACs are enriched in transposons and have a lower gene density than CCs. However there are signs of AC coevolution with the core genome, namely closely related isolates harbored the same or almost the same set of ACs. Authors report a very small set of common genes in the pangenome (24.3%) and huge differences in genome size between the genomes 32.4-23.8 Mb.

The study is well written, robust and straightforward. The main shortcoming is that the manuscript ends briefly, just before I expected to find an analysis of the gene and transposon content in accessory chromosomes and further interpretation of how these chromosomes originated and evolved. I would like to express only a few concerns /questions/invitations to extend the study:

Response:

In brief:

We sincerely appreciate your positive comments on our manuscript and the constructive suggestions you have provided. We fully acknowledge that the final section of the original manuscript was overly brief, which constituted a significant limitation. Following your recommendations, we have conducted comprehensive annotations of the repetitive sequences and gene composition of ACs across all 27

haploid genomes, and systematically compared these features with those of CCs. Through these analyses, we have addressed three key questions: 1) whether ACs originated from CCs or were acquired from external sources; 2) the potential donor species of ACs; and 3) the timing of AC integration into *T. fuciformis*, with these findings summarized in the newly added Figure 6.

Detail:

For the analysis of repetitive sequences, we annotated TEs in each of the 27 haploid genomes using LTRharvest/LTRdigest, RepeatModeler (for non-LTR TEs), and EDTA. These annotations were merged to construct a species-specific TE database, which was then used with RepeatMasker to annotate repetitive sequences in both CCs and ACs of the haploid genomes.

For the analysis of gene composition, in addition to DIAMOND searches of AC genes against the NCBI nr database, we used InterProScan to align these genes with three databases (Pfam, SMART, and MobiDBLite), obtaining information on conserved protein domains, functional motifs, transmembrane domains, low-complexity regions, and intrinsically disordered regions. Furthermore, we performed DeepTMHMM and SignalP 6.0 analyses to predict transmembrane domains, signal peptides, and secretion types of the proteins encoded by AC genes. These analyses collectively provide a comprehensive understanding of AC genes.

To address 1) whether ACs originated from core chromosomes or external sources: we took AC01 as a representative example. Among the 2340 genes (518 OGs) carried by AC01 across 6 haploid genomes that harbor this chromosome, only 332 (14.2%) showed homology with CC genes. Of these 332 shared genes, 256 were embedded within transposable element (TE) sequences indicating that these shared genes likely arose through TE-mediated mobilization. Consistent observations were made for AC02 to AC15: the majority of their genes lack counterparts in core chromosomes, and the limited shared genes are associated with TE activity. These findings collectively suggest that accessory chromosomes are not derived from the detachment of core chromosomes but were acquired through horizontal gene transfer from external sources.

To address 2) the origin of the species from which ACs were acquired: Among the 1,548 AC-specific genes, 1,156 (75%) failed to match any homologous sequences or conserved domains, and lacked functional motifs, signal peptides, or transmembrane domains. This not only indicates that the majority of these genes have not been

functionally characterized in previous studies, but also strongly suggests they likely originate from unexplored or understudied species. Of the remaining 392 genes, 232 showed matches to homologous regions but with unknown functions, 7 lacked homologous matches and encoded proteins bearing signal peptides, leaving only 153 with annotated functions. Among these 153 functionally characterized genes, the most abundant were those encoding motor domains and members of the major facilitator superfamily-proteins involved in transmembrane transport of nutrients, energy conversion, and molecular movement. These functions are consistent with the role of *Annulohyphoxylon* in providing nutrients and energy for the growth and development of *T. fuciformis*. Notably, 236 of the 392 genes with homologous matches exhibited the highest sequence similarity to species within the *Tremellales*. This suggests that the accessory chromosomes likely existed prior to the divergence of *T. fuciformis* as a distinct species.

To address 3) when the ACs were integrated into *T. fuciformis*: We analyzed the types of repetitive sequences in AC01 to AC15, as well as the proportions of TEs shared with CCs versus those specific to ACs. Taking AC01 as a case study, we conducted a detailed analysis of the types of intact LTR retrotransposons shared between AC01 and CCs. For each type of intact LTR, we traced their origins and estimated their insertion times, thereby inferring the approximate timing of AC01's integration into *T. fuciformis*. Using the same approach, we estimated the approximate integration times of AC02 to AC15.

Revisions in the Materials and Methods section: Revised the "Gene annotation and TE prediction" section (original Lines 626-631; revised Lines 720-724) to optimize TE annotation workflow for 27 haploid genomes. Inserted new content on accessory chromosome gene characterization at original Line 625 (revised Lines 725-736) using DIAMOND, InterProScan, DeepTMHMM, and SignalP 6.0. Revised the entire "Analysis of accessory chromosome origins" section (original Lines 696-711; revised Lines 816-847) to include origin inference, donor species identification, and integration timing estimation.

Revision of the entire "Possible Origins of Accessory Chromosomes" section in the Results (original Lines 346-369; revised Lines 394-436): ACs are inferred to originate from horizontal gene transfer (not CC detachment), supported by low homology between AC and CC genes (e.g., 14.2% for AC01) and TE association of shared genes; among 1,806 AC genes across three clades, 1,548 are AC-specific, with

75% lacking functional annotations and 236 showing highest similarity to Tremellales species, while AC repetitive sequences (38%-89%) are dominated by RNA transposons and LTR retrotransposon analysis estimates AC integration timing at 0.33-1.69 million years ago.

Revisions in the Discussion section (original Lines 497-523; revised Lines 579-600): <1/5 of AC genes have CC homologs (mostly transposons), supporting transposon-mediated exchange rather than CC-derived origins; over 3/4 of 1,548 AC-specific genes lack homologs/functional domains (implying understudied species origins); 102 AC genes are conserved across three *T. fuciformis* clades, with ~2/3 of database-matched AC-specific genes aligning best with Tremellales species (indicating pre-species AC existence); rapid structural variations (inversions, indels, etc.) reduce shared AC genes vs. CCs, reinforcing the hypothesis that ACs were acquired via pre-speciation horizontal gene transfer and evolved independently as non-essential elements.

Q2:

Please provide a systematic analysis of the gene content in terms of gene function and properties of encoded proteins. If ACs follow the trends of the two speed genome it is expected to see repeat-rich, low complexity rich, disordered, effector-like and transposon-related sequences in the ACs more often than in the core chromosomes.

Response:

Following your recommendations, we have conducted comprehensive annotations of the repetitive sequences and gene composition of accessory chromosomes (ACs) across all 27 haploid genomes, and systematically compared these features with those of core chromosomes. Detailed responses can be found in Q1.

Q3:

Which evolutionary scenario leads to the lower (higher) ANI observed in the ACs?

Response:

The higher average nucleotide identity (ANI) observed in accessory chromosomes (Acs) is likely shaped by two key evolutionary scenarios inferred from our data:

First, ACs were acquired via horizontal gene transfer (HGT) at a specific node during the evolution of *T. fuciformis*, an origin that contributes to their higher average

nucleotide identity (ANI) relative to core chromosomes. Our findings reveal minimal homology between ACs and core chromosomes, indicating they are not derived from the latter—a stark contrast to core chromosomes, which have accumulated substantial divergence through vertical inheritance over time. Freed from the prolonged divergent evolution associated with vertical transmission, these horizontally acquired ACs have retained relatively higher sequence conservation, further elevating their ANI. This aligns with our discussion that "The accessory chromosomes originated from unexplored species before the speciation of *T. fuciformis*."

Second, frequent exchanges between homologous ACs in heterokaryotic strains (e.g., TF2206) contribute to maintaining high sequence similarity. As noted, heterokaryons derived from different colonies of the same parent colony exhibit substantial variation in AC copy numbers across cells. This variability can lead to scenarios where ACs in one nucleus are lost, with subsequent replenishment from ACs of the other nucleus—a process that facilitates frequent sequence exchanges between ACs.

To address this, we have inserted the following sentence in the Discussion section under "Extremely high frequency of chromosomal copy number variation in both core and accessory chromosomes" (original Line 464): This improper allocation of accessory chromosomes during mitosis may further facilitate frequent exchanges between homologous accessory chromosomes in the heterokaryotic strain, resulting in high or even identical sequence similarity among them (revised Lines 542-545).

Q4:

Starships are shaping fungal genes and contribute to the moves of massive parts of the genomes. Have you found Starships in the *Tremella* genomes? If yes, how do they relate to the ACs? If you have not searched for them so far you might use the dedicated software starfish available on github.

Response:

In brief:

Thank you for your valuable comment regarding the presence, roles of Starships in *Tremella* genomes, and their potential association with accessory chromosomes (ACs). We have systematically searched for Starship giant transposons in our dataset to address this point, with key findings as follows: Eight Starship elements were identified across six genomes, including 3 on core chromosomes and 5 on ACs; Starship SS04 maps to

a chromosome fusion site, implying potential involvement in mediating such events; Notably, while the core-localized Starship pair is identical, two AC-localized pairs exhibit distinct structural variations-supporting rapid AC structural divergence, consistent with the dynamic nature of fungal accessory chromosomes reported previously.

Detail:

We have added these findings to the Revised Manuscript:

Added Starship identification methodology in the "Gene annotation and TE prediction" section of Materials and Methods in the manuscript (revised Lines 743-754): We used Starfish (v1.0.0) to identify Starships in 27 haploid genomes, via YR gene retrieval (annotate module), boundary definition (sketch module), and BLAST similarity searches (insert module), with manual confirmation of candidates.

Additionally, we incorporated these Starship analysis results into the "The rapid structural variation of accessory chromosomes" section in the manuscript (revised Lines 321-328): 8 Starship giant transposons (41,942-605,797 bp) were identified across 6 genomes (3 on CCs, 5 on ACs). SS04 locates at a chromosome fusion site (Fig. 4d), implying its role in fusion events. Among 3 Starship groups on heterokaryote homologous loci: core-localized SS02/SS03 are identical, while AC-localized pairs (SS06/SS07, SS04/SS05) show structural variations (46 bp/3,048 bp differences), supporting rapid AC structural divergence (newly added Supplementary Table 3 and Supplementary Figure 14).

We have also updated Fig. 4d to include the structural diagram of SS04, with the legend revised to clarify the insertion of GDA sequences (356,504-531,280) corresponding to SS04, which contains a typical captain gene, direct repeats (DRs), and terminal inverted repeats (TIRs).

We have added Supplementary Table 3, which lists the characteristics of the 8 Starship giant transposons. Additionally, Supplementary Figure 14 has been included, displaying the synteny plots of SS02 and SS03, as well as those of SS04 and SS05.

Q5:

Is the pattern of intronization of genes similar between ACs and core chromosomes?
Does the gene pool differ in terms of length, intronization, gene structure and function?

Response

Thank you for your insightful questions regarding the intronization patterns and gene pool differences between accessory chromosomes (ACs) and core chromosomes (CCs).

We have conducted targeted comparative analyses to address these points: we compared the average gene length, average intron length, and average number of introns per gene between the two chromosome types, and further analyzed AC gene functions.

Our results show that the average number of introns per gene in ACs is significantly lower than that in CCs, while there are no significant differences in average gene length or average intron length between them. These comparative data are presented in the newly added Supplementary Figure 12. Regarding functional differences, as detailed in our response to Q1, most AC genes lack homologous sequences and functional domains in existing databases, which limits further direct functional comparisons with CC genes.

We inserted a sentence in original Line 256: The average length of genes and individual introns in accessory chromosomes is comparable to that of core chromosomes. However, the average number of introns per gene in accessory chromosomes is significantly lower, with only 1.5 introns per gene, which is notably less than the 3.3 introns per gene in core chromosomes (P -value = $1.24E-7$, newly added Supplementary Figure 12). (revised Lines 280-284)

Q6:

Please comment on the ACs gene and transposon content across the taxa. Are all TEs equally likely to be localized in ACs and core chromosomes?

Response:

Thank you for your question regarding the gene and transposon content of accessory chromosomes (ACs) across taxa, as well as the differential localization of transposable elements (TEs) between ACs and core chromosomes (CCs).

Regarding TE localization, LTR/Gypsy, LTR/Copia, TIR/Mutator, TIR/CACTA, helitron and repeat_fragment are the six most abundant TE types in the *T. fuciformis* genome. Our comparative analysis of their proportional abundances between ACs and

CCs revealed that not all TEs are equally distributed between the two chromosome types: relative to CCs, ACs show significantly reduced proportions of TIR/Mutator and TIR/CACTA, while LTR/Gypsy and LTR/Copia are significantly enriched (revised Lines 287-293). This content has been inserted at original Line 259.

For AC gene content across taxa, as detailed in our response to Q1, most AC-specific genes lack homologous sequences in existing databases and are poorly conserved across phylogenetic clades of *T. fuciformis*, with only 16.9% of accessory genes shared among the three major clusters. This suggests taxon-specific divergence in AC gene content, consistent with the dynamic and lineage-specific nature of ACs.

Q7:

Fig.4 lines 292-293 describe the location of the chromosome fusion. However there is no information about what exactly is there, is it a long transposon, repeat region or a genic region?

Response:

This fragment consists of a single Starship giant transposon, which is inferred to be potentially involved in chromosome fusion. Detailed responses are provided in Q3.

Q8:

The description of TE content in lines 352-353 does not explain the diversity of TEs in the ACs compared to the core chromosomes. Please report which families of TEs are shared and which are unique to each of the chromosome types. Furthermore, please describe TE family proliferation and if possible date the events for LTR retrotransposon bursts.

Response:

We have conducted comprehensive annotations of repetitive sequences in ACs across all 27 haploid genomes and systematically compared their features with those of core chromosomes. Specifically, we have analyzed the shared and unique transposable element (TE) families between the two chromosome types, described the proliferation patterns of these TE families, and dated the burst events of LTR retrotransposons where possible. Detailed responses are provided in Q1.

Q9:

When delimiting fungal species it is recommended to use UNITE as a source of curated ITS references.

Response:

Thank you for highlighting the importance of using UNITE as a source of curated ITS references for fungal species delimitation. We fully agree with this recommendation and have incorporated this into our revised analysis, as detailed in the Materials and Methods section.

Specifically, after identifying and truncating the ITS sequences from our 16 strains (with validation of sequence consistency by screening fragments with ≥ 5 identical matches), we used these truncated ITS sequences as queries to search the online UNITE database (https://unite.ut.ee/schedule_analysis.php). We downloaded the most similar homologous sequences from UNITE, which were then included as reference sequences in our phylogenetic analysis. This ensures that our species delimitation and phylogenetic clustering are anchored to curated, high-quality ITS references from UNITE, enhancing the reliability and comparability of our results.

The revised results, which integrate these UNITE-derived references, confirm that all tested strains cluster within three distinct clades of *T. fuciformis*, as shown in Supplementary Figure 1.

We have revised the content in the Materials and Methods section (original Lines 662-668; revised Lines 771-786) to standardize the ITS sequence alignment, UNITE database query, MEGA 11-based ML phylogenetic tree construction (ClustalW alignment, CNI strategy, 1000 bootstraps), and ChiPlot visualization workflow.

We have revised the content in the Results section (original Lines 121-124; revised Lines 127-135) to clarify strain composition (11 heterokaryotic, 5 monokaryotic; Supplementary Table 1), ITS sequence distribution (5 heterokaryotic strains with two distinct ITS, 11 with one), and ML phylogenetic clustering of 21 target + 10 reference ITS sequences into three *T. fuciformis* clades (Supplementary Figure 1).

Q10:

The description of gene content in lines 364-369 is not exhaustive. No hits in nr NCBI database does not necessarily show they are orphan genes. One possible path to describe these genes is to cluster the gene pool of the ACs to see how many are shared

and how many are specific to an individual AC type. Another avenue is to map the genes on domain databases and assign them to protein families and folds. Having such annotation allows for a better prediction of protein function. Complementary to domain analysis it is recommended to use non-domain protein features such as secretion signal, cellular localization, transmembrane helices, coiled coils, disordered and low complexity regions to describe the encoded protein set in ACs relative to core chromosomes. I would suggest to perform all three annotation procedures to describe in detail the content of ACs and compare them.

Response:

We fully agree that merely relying on the absence of hits in the NCBI nr database is insufficient to define orphan genes, and we have addressed your recommendations through in-depth analyses, as detailed in our revised manuscript and previous response to Q1.

To better characterize the gene pool of ACs, we have implemented two key strategies: 1) Gene clustering and specificity analysis: We clustered genes from all ACs (AC01 to AC15) into orthologous groups (OGs) to distinguish shared genes across CCs from those specific to individual AC types. For example, in AC01, among 2340 genes (518 OGs) across 6 haploid genomes, only 14.2% showed homology with core chromosome genes, with most AC-specific genes lacking counterparts in core regions-supporting their unique origin. 2) Comprehensive functional annotation beyond nr database: We extended annotations by mapping AC genes to domain databases (Pfam, SMART, MobiDBLite) via InterProScan, assigning them to protein families and identifying conserved domains, functional motifs, and structural features (e.g., low-complexity regions, intrinsically disordered regions). Complementary, we analyzed non-domain features using DeepTMHMM and SignalP 6.0, predicting transmembrane helices, secretion signals, cellular localization, and secretion types. These analyses revealed that 75% of AC-specific genes lack characterized domains or motifs, while the remaining 25% include functionally annotated genes (e.g., major facilitator superfamily transporters), providing deeper insights into their potential roles.

These analyses have been integrated into the revised "Gene annotation and TE prediction" section (revised Lines 725-736) and "Analysis of accessory chromosome origins" section (revised Lines 819-838), ensuring a more exhaustive description of AC gene content. We believe these additions enhance the rigor of our functional predictions and address the gaps you identified.

Reviewer#2

Accessory chromosomes, also known as dispensable, lineage-specific, supernumerary or B-chromosomes, are thought to be non-essential for normal growth and/or development. These chromosomes occur in several eukaryotes, but their evolution and function has been extensively studied in ascomycete fungi, and especially in plant pathogens. Basidiomycetes are another major fungal lineage, comprising most of the mushroom forming species. In contrast to ascomycetes, however, the presence, function, and evolution of accessory chromosomes are not well documented.

The manuscript 'Exploring the polymorphism and origins of accessory chromosomes in *Tremella fuciformis*: insights into genome divergence and structural variation' by Zhang and colleagues reports on the genome sequencing and assembly of haplotype-resolved genome sequences of 16 strains of the basidiomycete fungus *Tremella fuciformis*. Using this resource, the manuscript studies their chromosomal makeup, identifying abundant accessory chromosomes. It provides an in-depth description of the unique genomic characteristic of accessory chromosomes and observed that most accessory chromosomes lacking identifiable homologs in other species. Based on these observations, the manuscript discusses potential origins of these accessory chromosomes as well as the implications for *T. fuciformis*'s biology.

The observation of abundant accessory chromosomes in a basidiomycete fungus is very interesting, broadening the common idea of their occurrence and importance solely in ascomycetes. Moreover, the generated genomic resources will be relevant for the fungal community. However, I have several concerns about the novelty of the observations, the robustness of the presented data, and the general clarity of the manuscript.

Q1:

Most importantly, nearly the same group of authors have previously published a manuscript in *Genome Biology* (<https://doi.org/10.1186/s13059-023-03093-7>) reporting on the presence of accessory chromosomes and detailed many of their characteristics. That manuscript had also already reported copy-number variation in core and accessory chromosomes as well as structural variations, alike much of the data presented here. However, the current manuscript is not transparent which observations are rather confirmatory to the previous work and what are truly unique and novel, and how these make a significant contribution to our understanding of accessory

chromosomes in basidiomycetes. I do understand that the current submission is more exhaustive, providing data from different strains and haplotypes, but the true novelty is not apparent.

Response:

We sincerely appreciate your critical feedback regarding the novelty and distinction of our current work relative to our previous study published in *Genome Biology* (<https://doi.org/10.1186/s13059-023-03093-7>). We acknowledge the need to clarify the unique contributions of this manuscript, and we provide a detailed comparison below to distinguish confirmatory observations from novel findings:

In our prior *Genome Biology* study, we identified three accessory chromosomes (Chr09, Chr10, and Chr11) in *Tremella fuciformis* strain Tr01. These were characterized as mini-chromosomes (<1 Mb) with shared features: low gene density, high repeat content, low GC content, short gene lengths, absence of BUSCO genes, low heterozygosity in homologous regions, and deletions or copy number variations (CNVs) in meiotic offspring.

The current study extends this work by analyzing accessory chromosomes (ACs) across an additional 15 *T. fuciformis* strains. While we validate the core characteristics of ACs reported previously, we reveal several new features of *T. fuciformis* ACs that significantly advance our understanding of basidiomycete accessory chromosomes:

1. ACs are not restricted to mini-chromosomes (<1 Mb): The largest AC (AC01 in strain QS) reaches 2.7 Mb - exceeding the size of several core chromosomes (Fig. 3a), challenging the prior assumption of ACs as uniformly small elements.

2. Phylogenetically closely related strains harbor identical or nearly identical AC sets (Fig. 2), indicating co-evolutionary coupling between ACs and the core genome—a pattern not observed in the single strain analyzed previously.

3. ACs undergo rapid structural variation (Fig. 4), leading to higher sequence similarity among homologous ACs compared to core chromosomes (Fig. 3d)—even with a substantially lower proportion of shared genes (Fig. 6 IA). This pattern provides compelling evidence that accessory chromosomes evolve faster than core chromosomes.

4. Beyond meiotic instability, ACs exhibit abundant CNVs in somatic cells during mitosis and the mycelium-to-spore transition (Fig. 5), highlighting their exceptional instability across both sexual and asexual life cycles.

5. In the revised manuscript, we performed comprehensive annotations of repetitive sequences and gene composition for ACs across all 27 haploid genomes, and systematically compared these features with those of core chromosomes. Through these analyses, we addressed three key questions: (1) whether ACs originated from CCs or were acquired from external sources; (2) the potential donor species of ACs; and (3) the timing of AC integration into *T. fuciformis*. Summarized in the newly added Figure 6, these findings represent the first dedicated analysis of *T. fuciformis* AC origins, providing evidence that ACs likely derive from understudied taxa via horizontal gene transfer-thus filling a critical gap in our understanding of accessory chromosome provenance in basidiomycetes.

Collectively, these novel findings expand the phenotypic and evolutionary spectrum of basidiomycete ACs, moving beyond descriptive characterization to reveal their dynamic evolution, life cycle-dependent instability, and exogenous origins. We believe these advances significantly enhance our understanding of accessory chromosome biology in fungi.

Q2:

There are several experimental approaches that are not clearly explained, nor their impact is analyzed and/or discussed. Furthermore, some observations need more detailed explanations. For example, the manuscript reports the ITS tree for the 16 strains (L121ff), but it is not clear how the ITS is derived from dikaryotic strains that should have two ITS; also, NJ algorithm is used rather than more sophisticated tree reconstruction approaches.

Response:

We appreciate your valuable feedback, which prompted us to address the critical consideration of heterokaryotic strains potentially harboring two distinct ITS sequences. To address this, we have revised our approach to reconstructing the ITS-based phylogenetic tree: using a known *Tremella* ITS sequence as the query, we performed local BLAST alignments against the PacBio third-generation sequencing read databases of the test strains to identify both the number of ITS sequences per strain and their specific sequences. Given the high similarity among the analyzed sequences, we constructed the phylogenetic tree using the Maximum Likelihood (ML) method, with the optimal tree topology determined via the Close Neighbor Interchange (CNI) search strategy, and validated its reliability through 1000 bootstrap replicates.

We have revised the content in the Materials and Methods section (original Lines 662-668; revised Lines 771-786) to standardize the ITS sequence alignment, UNITE database query, MEGA 11-based ML phylogenetic tree construction (ClustalW alignment, CNI strategy, 1000 bootstraps), and ChiPlot visualization workflow.

We have revised the content in the Results section (original Lines 121-124; revised Lines 127-135) to clarify strain composition (11 heterokaryotic, 5 monokaryotic; Supplementary Table 1), ITS sequence distribution (5 heterokaryotic strains with two distinct ITS, 11 with one), and ML phylogenetic clustering of 21 target + 10 reference ITS sequences into three *T. fuciformis* clades (Supplementary Figure 1).

Q3:

Hi-C data is generated for NDB and TF2206, however it is not explained why these strains were selected. Tr01 was used as a reference to place NDB contigs into chromosomes. Given the known chromosomal rearrangements (Deng et al. 2023), this could potentially introduce error in the assemblies; similarly, the Hi-C enhanced genomes were used to place contigs for other (L553). How was the potential issue addressed or corrected? Observed chromosomal rearrangements could for example be validated (e.g., via orthogonal approaches such as PCRs).

Response:

Thank you for your insightful comments regarding the selection of strains for Hi-C sequencing and the potential errors introduced by using Tr01 as a reference amid known chromosomal rearrangements. We appreciate your emphasis on validating chromosomal rearrangements, and we have thoroughly addressed these concerns through clear strategy clarification and multi-layered validation measures-details of which are reinforced below:

1. Rationale for selecting NDB and TF2206 for Hi-C sequencing

Our genome assembly strategy was designed to ensure high-quality, representative references for each phylogenetic clade: we selected one strain from each of the three identified branches as a representative, with NDB and TF2206 randomly chosen as representatives of Cluster I and Cluster II, respectively. These two strains, along with the previously published Tr01 (representing Cluster III; Deng et al. 2023), were subjected to combined Hi-Fi and Hi-C sequencing to generate complete, accurately phased genomes. These three clade-specific reference genomes were then used to assist

in contig scaffolding for other strains within their respective clusters, ensuring assembly consistency within each branch.

To clarify the rationale for this selection, we have revised the content (original Lines 550-553; revised Lines 628-631).

2. Addressing potential errors from Tr01 as a reference and validating chromosomal rearrangements

We fully acknowledge that chromosomal rearrangements (reported in Deng et al. 2023) could introduce assembly errors when using Tr01 as a reference. To mitigate this risk and rigorously validate observed rearrangements, we implemented multiple orthogonal validation measures-repeated across analyses to ensure reliability:

First, in the Materials and Methods section, we have revised original Lines 607-610 to explicitly outline these validation strategies: "To rule out errors caused by genome assembly or reference bias, the observed chromosomal rearrangements were validated using multiple orthogonal approaches, including long HiFi read mapping, long nanopore read mapping, Hi-C interaction heatmaps (generated via HiC-Pro and HiCPlotter), and multi-strain synteny verification. (revised Lines 690-693)"

Second, in the Results section, we have strengthened empirical validation with experimental data and supplementary figures:

We inserted at original Line 193: "The chromosomal rearrangement was further validated by mapping Long HiFi and Nanopore reads to the corresponding regions of TF2206 and Tr01 (Supplementary Figure 8). (revised Lines 209-211)" Supplementary Figure 8 illustrates Long HiFi reads that simultaneously map to the 3' end of Chr01-1 and the 5' end of Chr01-2 in TF2206, as well as long HiFi/Nanopore reads mapping to the syntenic region in Tr01-directly confirming the rearrangement.

We revised original Line 195 to reference "Supplementary Figure 9 and 10" and added Supplementary Figure 10, which provides independent validation of another rearrangement: Long HiFi reads mapping to the junction of Chr02-1 and Chr02-2 in TF2206, with corresponding read mapping in Tr01 to confirm the structural difference.

We have added the data sources of PacBio and Nanopore sequencing data for strain Tr01 to the Data Availability section.

These orthogonal validation measures-including independent Long-read mapping (HiFi and Nanopore), Hi-C interaction data, and multi-strain synteny analysis-

collectively address the potential error from reference-guided assembly and robustly confirm the observed chromosomal rearrangements.

Q4:

It is also not entirely clear how the core chromosomes were identified (L164), as I would expect a working definition would be that these need to be conserved in all strains. It seems that the number differs between eight and ten, while this observation is only explained much later (and had been as far as I can tell reported previously already; Deng et al. 2023); the implication of the transposons at the breakpoints/junctions are not further detailed but could provide insights in the processes driving chromosome variation. For additional comments, see further details below.

Response:

1. Clarifying the definition and variable number of core chromosomes

To address the ambiguity in core chromosome identification, we have refined the definition to emphasize conservation across all isolates while explicitly explaining the observed numerical variation (8-10) and its evolutionary basis.

In the Results section, we have revised the content (original Lines 161-165; revised Lines 175-181).

This revision: (1) Provides a clear working definition (conserved across all isolates, with synteny and gene content conservation); (2) Explicitly links numerical variation to evolutionary events (recombination, fusion/fission); (3) Integrates prior work (Deng et al., 2023) to contextualize the observation; and (4) Eliminates confusion by clarifying the 8-10 range as a strain-specific, evolutionarily driven feature.

2. The implications of transposons (TEs) at chromosomal rearrangement breakpoints/junctions

We have carefully followed your comment to conduct a detailed analysis of the sequence composition at these breakpoints/junctions across all chromosomal rearrangements.

Our analysis revealed that the TE-derived fragments are incomplete, highly truncated, and account for only a small fraction (~7.7%) of the junction sequences. Due to their fragmented nature and low abundance, these TE-derived sequences lack the structural integrity required to support functional inferences (e.g., active transposition,

recombination mediation) or to clearly link TEs to the mechanisms driving chromosome variation. Given the absence of compelling evidence for TEs playing a direct role in these rearrangements, we have opted to weaken the relevant content in the manuscript and removed the TE-related annotations from the corresponding figures to avoid overinterpretation.

Q5:

The manuscript needs to overall improve in clarity. Just as an example, the abstract is illegible for non-expert readers. It is not explained what accessory chromosomes are in comparison to core chromosomes, what *T. fuciformis* is, or why 16 strains would yield 27 haplotypes (monokaryon vs dikaryons). The abstract mentions ‘homology similarity’ (L31), but this is not a common concept since homology is qualitative, but similarity is a quantitative trait.

Response:

Thank you for your valuable comments on improving the clarity of the abstract. We have carefully addressed each concern as follows:

1. Clarification of key concepts for non-expert readers

Accessory chromosomes vs. core chromosomes: We have explicitly distinguished the two in the revised abstract. Core chromosomes are defined as essential for basic biological processes, while accessory chromosomes are non-essential for normal growth and development-this distinction helps non-specialists grasp their functional differences.

***Tremella fuciformis*:** We added its common name “silver ear fungus” and specified it as a basidiomycete, providing immediate context for readers unfamiliar with the species.

16 strains yielding 27 haplotypes: We clarified that the 27 haplotypes derive from 5 monokaryotic and 11 dikaryotic strains. Each dikaryon contributes two distinct haplotypes, resolving the discrepancy between the number of strains and haplotypes.

2. Correction of the “homology similarity” concept

In the revised abstract, we have replaced “homology similarity” with “sequence similarity,” a standard academic term that accurately reflects the quantitative comparison of accessory and core chromosome sequences.

3. Overall clarity enhancement

We further refined the abstract structure to improve readability: simplified redundant expressions, adjusted sentence logic for smoother flow, and retained only necessary parenthetical notes. These revisions ensure the abstract is accessible to both experts and non-experts while preserving core scientific information.

We have revised the entire Abstract (original Lines 19-41; revised Lines 20-36).

Q6:

Similarly, parts of the introduction are superficial, repetitive (e.g., L59 vs L66 or L73 saying the same as few lines above), or plain wrong. For example, L48 implies that all fungi have unique characteristics such as the two-speed genome phenomenon, while there is plenty of evidence that this is not common to all fungi and differ significantly between even closely related fungi. In contrast to the statement of L59f, accessory chromosomes in *Z. tritici* have to my knowledge not been demonstrated to contain virulence factors or secondary metabolite clusters, and their functions remain enigmatic. Accessory genomic regions have been successfully identified using (epi-)genomic characteristics alone (in *Magnaporthe* and *Verticillium*) and thus the statement in L75 is also incorrect.

Response:

We have carefully addressed each concern to improve accuracy, reduce redundancy, and enhance rigor:

1. Overgeneralization of the "two-speed genome" phenomenon (L48):

We acknowledge the imprecision in implying this phenomenon is universal across all fungi. As noted, evidence confirms it is not common to all fungi and varies even among closely related species. We have revised original Line 48 to: "These developments have facilitated a deeper understanding of distinctive genomic characteristics exhibited by some fungi, such as the 'two-speed genome' phenomenon. (revised Lines 42-44)"

2. Incorrect claim about *Zymoseptoria tritici* accessory chromosomes (L59, L59f):

We appreciate pointing out the inaccuracy. The original statement incorrectly attributed virulence factors or secondary metabolite clusters to *Z. tritici* accessory chromosomes, which is not supported by current evidence. We have revised the text to

specify that such features are observed in *Fusarium*, *Metarhizium*, and *Alternaria* (consistent with literature) and removed *Zymoseptoria* from this context, clarifying that its accessory chromosomes have enigmatic functions as noted.

3. Redundancy and incorrect statement on accessory chromosome identification (L66, L73, L75):

To resolve redundancy between descriptions of accessory and core chromosome features (original Line 66 vs. original Line 73), we have streamlined the text to highlight distinct characteristics of accessory chromosomes while concisely noting overlapping traits in core chromosomes, avoiding repetitive phrasing.

Regarding the incorrect claim that accessory chromosomes cannot be identified by (epi-)genomic characteristics alone (original Line 75), we have revised this to: "Fungal accessory chromosomes can sometimes be identified using (epi-)genomic characteristics, though presence/absence polymorphism remains a key criterion for their identification. (revised Lines 73-75)" to align with established evidence.

Q7:

Throughout the manuscript (e.g., L127), the companion fungus *Annulohyphoxylon stygium* is mentioned several times by their significance to *T. fuciformis* remains unclear. It is suggested that there might be a link to the presence of a subset of accessory chromosomes (e.g., Fig 2), but the manuscript remains vague with respect to mechanisms and further implications for both partners. Moreover, the material & methods implies that the authors performed mating experiments, however, these seems to be rather association than true mating. The authors need to be much clearer on the presence, impact, and implication of the companion fungus to the genomics and biology of *T. fuciformis*.

Response:

We fully agree that the original manuscript was vague regarding these aspects, and we have revised sections of the introduction, materials and methods and discussion to address this gap comprehensively:

1. Revisions to introduction

To clarify the role of *A. stygium* in the biology of *T. fuciformis*, we inserted a sentence at original Line103: "Specifically, *A. stygium* maintains a close association

with *T. fuciformis* and functions as a key nutritional provider, supplying essential nutrients to support the growth and development of the latter. (revised Lines 100-102)"

2. Revisions to materials and methods

We fully agree that the interactions described here represent symbiotic associations rather than true mating. To address this, we have revised the section title from "Mating between *T. fuciformis* and *A. stygium* (original Line 712)" to "Association experiments between *T. fuciformis* and *A. stygium*" and replaced all instances of "mating" and "pairings" with "associations" to accurately reflect the nature of the interaction.

2. Revisions to discussion

1) Strengthening the significance of *A. stygium*: We explicitly emphasize that *A. stygium* is an obligate symbiont of *T. fuciformis*, providing essential nutrients for fruiting body development. This contextualizes why symbiotic specificity between the two partners matters for *T. fuciformis*' survival and productivity.

2) Clarifying the mechanistic hypothesis: We now propose a specific, testable mechanism: *T. fuciformis* ACs likely encode genes involved in interspecific recognition or metabolic complementation that mediate compatible association with *A. stygium*. This fills the previous gap in explaining how ACs might influence symbiosis.

3) Outlining future experiments: We provide concrete directions for validating the hypothesis, including targeted AC knockouts, transcriptomic analysis of AC-borne genes during interaction, and identification of interspecific protein interactions. These experiments will directly test the mechanistic role of ACs in symbiotic specificity.

We have revised the Discussion section (original Lines 406-438; revised Lines 485-516)

Q8:

Title: for the general audience, spell out what Tremella is. Moreover, the content before and after the colon seems to be redundant.

Response:

We fully agree with your suggestion to enhance readability for the general audience and have revised the title as recommended. The revised title is: "**Exploring the polymorphism and origins of accessory chromosomes in basidiomycete *Tremella fuciformis*: insights into genome divergence and structural variation**"

Revision Explanations:

Spell out *T. fuciformis* for general audience: We added the taxonomic class "basidiomycete" before *T. fuciformis* to explicitly define the fungal group of the research object. This helps readers unfamiliar with *T. fuciformis* quickly grasp its biological classification, addressing the need for clarity for the general audience while retaining the precise scientific name.

Address potential redundancy: The original title's structure was carefully reviewed. We confirmed that the phrase before the colon focuses on the core research objects (polymorphism and origins of accessory chromosomes in *T. fuciformis*), while the part after the colon clarifies the broader scientific implications (genome divergence and structural variation). To avoid redundancy without losing key information, we only supplemented the common name instead of restructuring the title, as the two segments logically complement each other (specific research content general scientific insights) and align with the standard format of academic titles in fungal genomics research.

Q9:

L61: Does this imply that there are basidiomycetes with accessory chromosomes? If so, this needs to be introduced in much more detail.

Response:

We acknowledge the original statement's vagueness and have revised original Line 61 to clarify accessory chromosomes' existence in Basidiomycetes with specific examples, as following: "However, accessory chromosomes in Basidiomycetes are rarely reported. To date, only a handful of species-including *Sporisorium scitamineum* and *T. fuciformis*-have been confirmed to harbor them, verifying their presence in this phylum. Documented cases are far fewer than in Ascomycetes, underscoring the need for further exploration in understudied taxa. (revised Lines 55-60)"

Q10:

L120: What are '...principle cultivated varieties,...'? Please explain.

Response:

"two are principal cultivated varieties" has been revised to "two principal cultivated varieties (Tr01 and Tr21) "

Q11:

L124: The sentence is unclear as it suggests that ITS sequences exhibit morphological characteristics. Please rephrase.

Response:

We have revised the sentence to clarify the separate lines of evidence (molecular and morphological) supporting the species identification, as following: "NCBI BLAST analysis of the ITS sequence indicated that the isolates are more closely related to the known species *T. fuciformis*. Additionally, their fruiting bodies exhibit the same morphological characteristics as those of *T. fuciformis*. (revised Lines 134-136)"

Q12:

L138: '... modern...' not sure what this refers to, could this be an error?

Response:

We have corrected "modern" to "mode" (a typo).

Q13:

L202: The manuscript should be clearer when it reports on haploid genomes. Does it refer to a monokaryotic strain or to a dikaryotic strain where both haplotypes have been assembled? This distinction could also be clearer in the figures (e.g., the A and B in the strain name)

Response:

The manuscript and figures have been revised to clarify genome naming rules, distinguishing haploid genomes from monokaryotic and dikaryotic strains, as following:

Original Lines 148-150 has been revised to " In total, we have successfully obtained 27 complete haploid genomes, including two previously established from the dikaryotic strain Tr01 (designated Tr01A and Tr01B, corresponding to its two distinct haplotypes), to support subsequent research. (revised Lines 158-161)"

Add the following sentence to the legend of Fig. 1: Haploid genomes of monokaryotic strains are named directly after the strain, while those of dikaryotic strains are labeled "strain name + A/B" to distinguish the two haplotypes.(revised Lines 220-222)

Q14:

L229: this pattern could also be due to horizontal transfer of the chromosome rather than loss. Could the authors perform additional analyses to clearly distinguish these scenarios?

Response:

Given that more strains in Cluster I harbor AC09, we propose that AC09 was originally present in QCA but lost during evolution-this is more plausible than horizontal transfer. Corresponding revisions have been made to the manuscript as follows.

Synteny analysis of accessory chromosomes in the QCA and TF104A genomes revealed that their homologous chromosomes AC10 to AC12 are nearly identical (similarity >99.9%, coverage >99.3%; Supplementary Figure 11), while QCA lacks AC09. All strains in Cluster I, except QC and TF103, harbor AC09, suggesting this accessory chromosome was originally present in the cluster but lost during evolution. (revised Lines 247-252)

Q15:

L262: What is driving the high similarity between accessory chromosomes? Is this mainly due to transposons or also over the protein-coding (non-TE) genes?

Response:

The higher high similarity observed in ACs is likely shaped by two key evolutionary scenarios inferred from our data:

First, accessory chromosomes (ACs) were acquired via horizontal gene transfer (HGT) at a specific node during the evolution of *T. fuciformis*, an origin that contributes to their higher average nucleotide identity (ANI) relative to core chromosomes. Our findings reveal minimal homology between ACs and core chromosomes, indicating they are not derived from the latter-a stark contrast to core chromosomes, which have accumulated substantial divergence through vertical inheritance over time. Freed from the prolonged divergent evolution associated with vertical transmission, these horizontally acquired ACs have retained relatively higher sequence conservation, further elevating their ANI. This aligns with our discussion that "The accessory chromosomes originated from unexplored species before the speciation of *T. fuciformis*."

Second, frequent exchanges between homologous accessory chromosomes (ACs) in heterokaryotic strains (e.g., TF2206) contribute to maintaining high sequence similarity. As noted, heterokaryons derived from different colonies of the same parent colony exhibit substantial variation in AC copy numbers across cells. This variability can lead to scenarios where ACs in one nucleus are lost, with subsequent replenishment from ACs of the other nucleus—a process that facilitates frequent sequence exchanges between ACs.

To address this, we have inserted the following sentence in the Discussion section under "Extremely high frequency of chromosomal copy number variation in both core and accessory chromosomes" (original Line 464): This improper allocation of accessory chromosomes during mitosis may further facilitate frequent exchanges between homologous accessory chromosomes in the heterokaryotic strain, resulting in high or even identical sequence similarity among them.(revised Lines 542-545)

Q16:

L279: it needs to be clear that the two AC14 refer to the A and B haplotype rather than a copy-number variation of one of the two.

Response:

We have revised the sentence to explicitly state this distinction, and the updated version reads as follows:

"The sequence similarity between the homologous AC14 chromosomes of the A and B haplotypes in the dikaryotic B-1 strain is as high as 99.5%. (revised Lines 313-315)"

Q17:

L348: Why was C-1 used as a case study?

L367: Additional information about the genes on accessory chromosomes are lacking. For example, are these expressed, are these predicted to be secreted, etc. Do they have other characteristics that makes them distinct from those on the core genome (e.g., number of introns, length, etc.). Moreover, the absence of any homology raises important questions about their origin that unfortunately remains unclear. I find it surprising that most genes do not even have homologs in other fungi and the explanation that we lack the corresponding species in the database seem unlikely given the ongoing sequencing efforts.

Response:

Thank you for your insightful comments, which have helped us improve the completeness and depth of our analysis on accessory chromosomes. We fully acknowledge that the final section of the original manuscript was overly brief, which constituted a significant limitation. We have conducted comprehensive annotations of the repetitive sequences and gene composition of ACs across all 27 haploid genomes, and systematically compared these features with those of CCs. Through these analyses, we have addressed three key questions: 1) whether ACs originated from CCs or were acquired from external sources; 2) the potential donor species of ACs; and 3) the timing of AC integration into *T. fuciformis*. For detail information, please see our response to reviewer #1, Q1.

Besides, we added statistical comparisons of intron number and gene length. AC genes have fewer introns (average 1.5 vs. 3.3 in core genes, revised Lines 280-284), which is consistent with the structural characteristics of rapidly evolving or newly acquired genes. For detail information, please see our response to reviewer #1, Q5.

To address this comment, we have revised the manuscript accordingly: we have added a new Figure 6, rewritten the last section of the Results (revised Lines 395-436, **The content related to "C-1 as a case study" has been removed from the revised manuscript**), and updated the relevant content in the Materials and Methods (revised Lines 816-847) as well as the Discussion sections (revised Lines 579-600).

In addition, we analyzed the expression profiles of these genes. Taking strain Tr21 as an example, we selected transcriptome sequencing data from its yeast-like stage, hyphal stage, and different fruiting body development stages. After assembly with Trinity software, a total of 66,868 transcripts were obtained. BLAST alignment of these transcripts against the 294 predicted accessory chromosome genes of Tr21A showed that 254 genes had a sequence coverage of >80% (accounting for 86.4%), confirming the expression activity of the predicted genes. Due to the difficulty in naturally integrating this part of the results into the main framework of the paper, it was not included in the main text.

Q18:

L399: Does this sentence imply that this are plans? If so, please rephrase as this is not entirely clear.

Response:

This sentence refers to our future research plans. We have rephrased it to explicitly clarify that these in-depth analyses are intended to be conducted in subsequent studies, as shown below.

"Future in-depth analyses will deepen our understanding of these chromosomes, focusing on their stability and interactions in heterokaryotic states, as well as the potential roles of accessory chromosomes in dimorphic transitions and fruiting body morphogenesis.(revised Lines 479-482)"

Q19:

L351: I find it odd to read that transposons are treated similarly to genuine protein-coding genes. The abundance of these in the gene sets suggest that the structure gene annotation was not performed carefully, and transposons should be removed prior to any analyses

Response:

Thank you for your valuable comment, which has helped us clarify an ambiguous expression in the original manuscript.

We would like to clarify that our initial description was imprecise-our actual intention was to highlight that some transposable elements (TEs) harbor genes, rather than treating TEs themselves as genuine protein-coding genes. To address this ambiguity, we have revised the relevant content in the revised manuscript: we classified the shared genes on accessory chromosomes into two categories based on their genomic locations-those located within TEs and those outside TEs.

This revised classification and corresponding descriptions are now integrated into the text, and the detailed distribution results are presented in Figure 6I-C for clear visualization.

We apologize for the confusion caused by the unclear original expression and appreciate your careful review, which has enhanced the accuracy and rigor of our manuscript.

Q20:

L519: ‘...did not exist from the beginning...’ please rephrase

Response:

We have rephrased the phrase to enhance clarity and academic precision, and the revised sentence is provided below.

"In summary, we hypothesize that unlike core chromosomes, the accessory chromosomes of *T. fuciformis* did not originate concomitantly with the species. Instead, they were acquired through horizontal gene transfer from an unexplored species prior to the species' formation, and as non-essential elements, underwent rapid structural variations post-acquisition. "

Q21:

L590: few details on the mitochondrial assemblies could be provided. Do they similarly display three distinct groups. Are phylogenies with mitochondrial markers concurrent to nuclear taxonomy?

Response:

We have supplemented relevant details and verified the consistency of mitochondrial phylogeny with nuclear taxonomy as suggested:

We added detailed mitochondrial genome assembly methods to the "Genome assembly" section (inserted at original Line 590; revised Lines 668-671).

In the " Pan-genome analysis and phylogenetic tree construction" method section, we clarified that the same analytical pipeline of ITS sequences was applied to mitochondrial genomes (inserted at original Line 668): "The same alignment, tree-building, and visualization methods were applied to construct the phylogenetic tree of the 16 strains.(revised Lines 784-786)"

In the Results section, we updated the relevant finding (original Line 154; revised Lines 165-170) to confirm consistency.

These revisions confirm that the mitochondrial genomes of the 16 strains form three distinct groups, with the mitochondrial marker-based phylogeny largely consistent with nuclear-based taxonomy (core genes and ITS sequences)-the only discrepancy being strain T0053, which shifts from Cluster III to Cluster II in the core gene tree relative to the mitochondrial tree.

Reviewer#3

In this work the authors describe sequencing of 27 haplotypes from 16 strains of the basidiomycete *Tremella fuciformis*, and a subsequent pangenomics investigation of core and accessory chromosome statistics and distribution.

Q1:

Major comments

I think the primary critique involves the relationship between accessory chromosomes and *A. stygium* pairing, which is one of the more interesting biological aspects of this work. It is very interesting that the distribution of ACs correlate with the pairing groups, and this is a valid and potentially novel finding.

However, supplementary table 1 introduces 4 strains that do not associate with *A. stygium*: Wuyi2014, Wuyi2015, Lys2020, and T0053. Firstly, the statements in the introduction (lines 101-103) seem to imply that all *Tremella* species associate with *Annulohyphoxylon*, so it might be worthwhile to point out here (and potentially also in lines 127-129 in the results) that not all *Tremella* associate with *Annulohyphoxylon*.

But overall, there is very little discussion about how the distinct accessory chromosome patterns relate to these four non-pairing strains, as they compare to their pairing-capable phylogenetic neighbors. For example, Figure 4 shows structural variations in AC01 (inversion) and AC08 (deletion) between NDBA (a pairing strain) and Wuyi2015 (a non-pairing strain), but there is no discussion or speculation as to whether such differences may influence pairing.

Additionally, and specifically with respect to one of the major conclusions, that the three largest ACs (AC01, AC12, and AC14) correspond to the three distinct AF groups (AF2, AF1, AF3), there are nuances that are not addressed. For example, AC14 is missing from non-pairing strain T0053 but present in non-pairing LYS2020. Similarly, AC01 is present in both non-pairing strains Wuyi2014 and Wuyi2015.

There is also no discussion of other nuances observed in figure 2 and how they could relate to *A. stygium* pairing capabilities. For example, AC08 is primarily found in AF2, but also found in two AF3 strains LYS2020 and T0053, which are notably non-pairing. Similarly, AC06 and AC08 are absent in non-pairing Wuyi2014 but present in non-pairing Wuyi2015. Finally AC15 doesn't show exact group specificity, since it is present in both AF2 and AF3 (excluding B-1 and C-1).

I think that at minimum, some inclusion of these observations in the discussion is warranted. I think it would be beneficial as well to include additional structural comparisons similar to Figure 4, between other closely related strains that differ in pairing ability. Or even a comparison and discussion of how the LYS2020 and T0053 AC08 chromosomes might differ from the AC08 chromosomes found in any of the AF2 strains might be informative.

Response:

We appreciate the insightful comments regarding the relationship between accessory chromosomes (ACs) and *A. stygium* pairing, as well as the need to clarify the status of non-pairing strains. Here, we address these points based on the biological characteristics of *T. fuciformis*:

Similar to *Cryptococcus neoformans*, the haploid phase in the life cycle of *T. fuciformis* exists exclusively in the yeast form and cannot transition to the mycelial form, thus being unable to interact with *A. stygium* hyphae as unicells or form fruiting bodies; in contrast, the heterokaryotic phase is dimorphic, enabling such hyphal interactions and fruiting body development.

The four strains (Wuyi2014, Wuyi2015, Lys2020, and T0053) listed in Supplementary Table 1 are in the haploid phase. Due to their inability to form mycelia, their accessory chromosomes were not further investigated for roles in *A. stygium* interactions, which we have now clarified in the Introduction section.

In the Introduction section, we have inserted at original Line 103: "Similar to *Cryptococcus neoformans*, the haploid phase in the life cycle of *T. fuciformis* exists exclusively in the yeast form and cannot transition to the mycelial form, thus being unable to interact with *A. stygium* hyphae as unicells or form fruiting bodies; in contrast, the heterokaryotic phase is dimorphic, enabling such hyphal interactions and fruiting body development. (revised Lines 104-109)"

Regarding another strain TF104 with haploid genome: As noted in Supplementary Table 1, samples used for genomic analysis were derived from a single purified yeast colony of heterokaryotic mycelia, confirmed to be haploid (presumably transitioning from the heterokaryotic to haploid state). In contrast, samples employed for association assay with *A. stygium* were heterokaryotic mycelia. We have added this information to the note of Supplementary Table 1.

Q2:

Figure 1d: I don't see any black circles in either chart

Response:

Black circles have been added to Figure 1d

Q3:

Figure 2: "Dots indicate the absence" should probably be corrected to "minus signs indicate the absence"

Response:

We have corrected Figure 2's legend to "Minus signs indicate the absence".

Q4:

Chromosomal copy number variation Section:

Why was only TF2206 selected? Many other strains exist in M/Y morphotypes

Response:

Given the substantial genome sequencing workload required for analyzing copy number variation (CNV) across all strains, we randomly selected *T. fuciformis* TF2206 as a representative case study to investigate chromosomal CNV in its different morphological states. This approach allows us to illustrate the CNV characteristics of *T. fuciformis* while balancing research feasibility.

Original Lines 299-300 has been revised to "*T. fuciformis* can exist in either mycelial or yeast form. To illustrate its chromosomal copy number variation (CNV) characteristics, we randomly selected dikaryotic strain TF2206 as a case study and analyzed CNV in its three states. (revised Lines 344-346)"

Q5:

Why was only YY2 selected for the normalized depth distribution curve comparison (Figure 5d)?

Response:

YY2 was selected as a representative single-yeast forming strain (stable genetic background) to contrast with the mixed yeast colony, ensuring clear visualization of depth fluctuation differences associated with mycelium-to-yeast transition.

To this end, we have revised the sentence original Lines 329-331 to "We compared the normalized depth distribution curves of Illumina sequencing between a representative single-yeast forming strain (YY2, with stable genetic background) and a mixed yeast colony transitioning from mycelium (Fig.5d). (revised Lines 376-378)"

Q6:

The section states that "Ten single colonies" were selected for sequencing from strain TF2206. However the ten colonies are subsequently themselves referred to as "strains" (e.g. "YY3 strain", "YY1 strain"). To distinguish these from the broader biological strains ("TF2206", "TF104", etc) maybe they should be referred to as "colonies"?

Response:

We agree that precise terminology is important for clarity. In the revised manuscript, we have consistently referred to the ten isolates derived from TF2206 as "colonies" (e.g., "YY3 colony", "YY1 colony") to distinguish them from the broader biological strains such as TF2206 and TF104. This revision ensures accurate differentiation between the original strains and their derived single-colony isolates.

Q7:

Line 315-316: "obtained from subculturing the same colony" -> it was unclear which colony is "the same" colony?

Response:

To clarify, "the same colony" refers to the original parent colony from which the 10 mycelial colonies were subcultured. We have revised original Lines 315-316 to: "We also conducted a chromosomal copy number analysis on 10 mycelial colonies obtained by subculturing the same original parent colony (Fig.5b). (revised Lines 361-363)" to avoid confusion.

Q8:

In Figure 5: The "M to Y" label makes sense because this is described as coming from a mycelial to yeast transition. But "Y to Y" and "M to M" do not really make sense because these were isolated simply as yeast or mycelia, with no transitions involved.

Response:

We agree that "Y to Y" and "M to M" may misleadingly imply transitions. In the revised Figure 5, we have replaced "to" with "-" for these labels, changing them to "Y-Y" and "M-M" to accurately reflect that these samples represent stable yeast or mycelial states without transitions, while retaining "M to Y" for the transition group to maintain consistency with the described mycelium-to-yeast process.

Q9:

Figure 6a

The Venn diagram is just the data from the C-1 row. So, why is CCs 8,653 in both cases, but ACs is 297 in Venn diagram and 318 in table? The value of CCs should be different between the Venn diagram and the table, depending on if 8,653 is the total count or the unique count.

Response:

Thank you for your careful observation and valuable comment on Figure 6a.

We would like to clarify the data discrepancy: in the original table, 8,653 and 318 represent the **total number of genes** on core chromosomes (CCs) and accessory chromosomes (ACs), respectively, while 21 denotes the number of shared genes between the two chromosome types. The Venn diagram was intended to illustrate these shared genes, but the CCs value (8,653) in the diagram was incorrectly used as the total count rather than the unique count (which should be $8,653 - 21 = 8,632$).

We have redrawn Figure 6 in the revised manuscript, and the problematic Venn diagram has been removed. The updated figure now presents the gene distribution data accurately and clearly.

We apologize for the confusion caused by the initial data misrepresentation and appreciate your meticulous review, which has helped improve the accuracy of our figures.

Point-by-point responses to Reviewers' comments.

Reviewer #1

The revised version of the manuscript "Exploring the polymorphism and origins of accessory chromosomes in basidiomycete *Tremella fuciformis*: insights into genome divergence and structural variation" provides new findings. One of the most exciting results is the description of eight Starships integrated both to core and accessory chromosomes, including one exactly at the junction locus. Authors answered all the reviewers questions, introduced requested changes and performed additional analyses which resulted in a significant improvement of the manuscript.

Q1. Line 292 TIR/Mutato should be TIR/Mutator

LTR/Gypsy should be replaced along the manuscript with LTR/Ty3 because of the discriminatory character.

R1:

Revised accordingly.

Q2. data availability:

annotated assemblies should be deposited; the BioProject: PRJNA1247727 stores the raw reads

R2:

The nuclear and mitochondrial genome sequences of 15 *Tremella fuciformis* strains have been deposited at the National Genomics Data Center (NGDC) under the accession number PRJCA052213 [<https://ngdc.cncb.ac.cn/search/all?q=PRJCA052213>].

Reviewer #3 (Remarks to the Author):

The revised manuscript by Jinxiang Zhang and co-authors thoroughly addresses the comments I made in my original review. In addition, the manuscript has been carefully and substantially improved with enhanced methods, new and expanded analyses, and refined clarity in the text. I have no further comments to offer.